# Antibody and DNA sensing pathways converge to activate the inflammasome during primary human macrophage infection

Larisa I Labzin[1] (iD), Maria Bottermann[1], Pablo Rodriguez-Silvestre[1], Stian Foss[2], Jan Terje Andersen[3,4], Marina Vaysburd[1], Dean Clift[1] & Leo C James[1,*] (iD)

## Abstract

Inflammasomes are potent innate immune signalling complexes that couple cytokine release with pro-inflammatory cell death. However, pathogens have evolved strategies to evade this cell autonomous system. Here, we show how antibodies combine with innate sensors in primary human macrophages to detect viral infection and activate the inflammasome. Our data demonstrate that antibody opsonisation of virions can activate macrophages in multiple ways. In the first, antibody binding of adenovirus causes lysosomal damage, activating NLRP3 to drive inflammasome formation and IL-1β release. Importantly, this mechanism enhances virion capture but not infection and is accompanied by cell death, denying the opportunity for viral replication. Unexpectedly, we also find that antibody-coated viruses, which successfully escape into the cytosol, trigger a second system of inflammasome activation. These viruses are intercepted by the cytosolic antibody receptor TRIM21 and the DNA sensor cGAS. Together, these sensors stimulate both NLRP3 inflammasome formation and NFκB activation, driving dose-dependent IL-1β and TNF secretion, without inducing cell death. Our data highlight the importance of cooperativity between multiple sensing networks to expose viruses to the inflammasome pathway, which is particularly important for how our innate immune system responds to infection in the presence of pre-existing immunity.

**Keywords** adenovirus; antibody; inflammasome; macrophage; TRIM21
**Subject Category** Immunology
**The EMBO Journal (2019) 38: e101365**

See also: **RC Coll** (November 2019)

## Introduction

Macrophages are a key component of cellular immunity, acting as both infection sensors and anti-microbial effectors, with diverse sub-types found in tissues and draining lymph nodes throughout the body. Macrophages sense pathogens and actively ingest them via phagocytosis causing pathogen destruction in the phagolysosome, facilitating antigen presentation to B and T cells, and importantly releasing pro-inflammatory mediators to recruit other immune cells. However, the anatomical distribution and fundamental function of macrophages as pathogen sensors place macrophages themselves at a high risk of infection. Indeed, despite being exquisitely designed to suppress infection, macrophages can be colonised by many pathogens—a dichotomy referred to as the "macrophage paradox" (Price & Vance, 2014), and used as a means to disseminate the pathogen around the body (Klepper & Branch, 2015).

Bacteria are the most comprehensively studied of macrophage pathogens; however, viruses are also capable of infecting and replicating in macrophages. Influenza strains can replicate in mouse bone marrow-derived macrophages (Chan *et al*, 2012) and human monocyte-derived macrophages (HMDM; Hoeve *et al*, 2012). Highly pathogenic H5N1 influenza virus replicates in both dendritic cells (DCs) and macrophages (Westenius *et al*, 2018), while pandemic HIV-1 replicates efficiently in primary human macrophages (Rasaiyaah *et al*, 2013) as does HSV-1 (MacLeod *et al*, 2013). Macrophages are reported to have less acidic endosomes than epithelial cells (Marvin *et al*, 2017; and references therein), potentially hampering infection as viruses use low pH environments as cues to trigger fusion or membrane permeabilisation. However, viruses like adenovirus (AdV) do not require acidification and use mechanical cues to induce uptake and endosomal escape (Greber, 2016). Indeed, AdV specifically targets alveolar macrophages through the scavenger receptor SR-A6 (MARCO; Stichling *et al*, 2018). Taken together, the potential for viral infection suggests that macrophages need mechanisms that promote viral capture and immune detection but inhibit viral replication.

1   Protein and Nucleic Acid Chemistry Division, Medical Research Council, Laboratory of Molecular Biology, Cambridge, UK
2   Centre for Immune Regulation (CIR), Department of Biosciences, University of Oslo, Oslo, Norway
3   CIR and Department of Immunology, Oslo University Hospital Rikshospitalet, Oslo,Norway
4   Department of Pharmacology, Institute of Clinical Medicine, Oslo University Hospital, University of Oslo, Oslo, Norway
    *Corresponding author. Tel: +44 01223 267162; E-mail: lcj@mrc-lmb.cam.ac.uk

One mechanism to deny pathogens a niche to replicate in but simultaneously promote immune signalling is inflammasome-mediated cell death (Jorgensen *et al*, 2017). Inflammasomes are cytosolic multimeric signalling platforms which cleave the potent pro-inflammatory cytokines IL-1β and IL-18 into their mature forms and can trigger pro-inflammatory cell death (pyroptosis) via cleavage of the pore forming protein Gasdermin D (Lamkanfi & Dixit, 2014). Inflammasomes consist of a scaffolding "sensor" protein whose oligomerisation is induced by a pathogen-associated molecular pattern (PAMP) (e.g. Flagellin; NLRC4) or an endogenous danger signal (e.g. ATP; NLRP3), which subsequently recruit the adaptor protein ASC and the protease caspase-1. Canonical inflammasome activation is tightly regulated and typically restricted to the myeloid compartment, with NFκB signalling required to induce expression of pro-IL-1β and some components, e.g. NLRP3. This first NFκB signal is referred to as licensing as it is necessary but insufficient for inflammasome activation (Shim & Lee, 2018). A second trigger is required to induce oligomerisation of the scaffolding protein and recruit the adaptor proteins to form the active, multi-protein signalling complex. Once the inflammasome is formed, caspase-1 cleaves itself to self-activate, whereupon it can then cleave IL-1β, IL-18 and GSDMD. The N-terminal fragment of GSDMD then assembles to make pores in the plasma membrane eventually resulting in pyroptosis, a lytic cell death (Jorgensen *et al*, 2017). Though IL-1β can be released through GSDMD pores, recent studies have indicated that pyroptosis and even GSDMD pores themselves are not absolutely required for IL-1β cytokine release, as this can occur from living cells, such as neutrophils (Chen *et al*, 2014) and macrophages (Evavold *et al*, 2018) and in a GSDMD-independent manner (Monteleone *et al*, 2018).

Many viruses have been shown to activate the inflammasome during infection (Chen & Ichinohe, 2015). AdV has been the subject of many such studies due to its prevalence in the human population, capacity to cause fatal infection in immunocompromised individuals (e.g. during transplants) and because it is the most commonly used viral vector in gene therapy. In all these studies, incoming virions of the Ad5 serotype are being sensed, as inflammasome activation occurs in response to replication-deficient adenoviral vectors (Muruve *et al*, 2008; Zaiss *et al*, 2009; Barlan *et al*, 2011b; Teigler *et al*, 2014; Eichholz *et al*, 2016). However, multiple conflicting mechanisms and pattern recognition receptors (PRRs) have been proposed to activate the inflammasome during AdV infection. This is likely due to characterisation of this pathway in a multitude of cell types, including monocytes, macrophages and dendritic cells, and in human or mouse cell lines. As recent studies have shown, inflammasome pathways are not completely conserved between mouse and human, nor between cell types. For instance, the AIM2 inflammasome pathway which senses cytosolic DNA in mouse macrophages is not utilised in human monocytes and macrophages (Gaidt *et al*, 2017). Nevertheless, AdV has been reported to activate the inflammasome via NLRP3 (Muruve *et al*, 2008; Zaiss *et al*, 2009; Barlan *et al*, 2011b) or AIM2 (Eichholz *et al*, 2016) with additional NFκB-dependent priming of pro-IL-1β and NLRP3 itself via TLR9 (Barlan *et al*, 2011b; Teigler *et al*, 2014; Eichholz *et al*, 2016). What viral PAMPs (pathogen-associated molecular pattern) are required to drive inflammasome activation during infection is also unclear. Empty adenoviral capsids do not activate the inflammasome, suggesting viral genomes are the sensed ligand (Muruve *et al*, 2008; Eichholz *et al*, 2016). However, the Ad5 capsid is known to protect the genome from DNA sensing in non-myeloid cells (Watkinson *et al*, 2015).

Here, we show that adaptive immunity (antibodies) combines with innate immune sensors to reveal AdV to the inflammasome in primary human cells. We show that antibody-dependent IL-1β release is NLRP3 dependent, and we uncover a novel role for TRIM21 upstream of NLRP3 and cGAS, resulting in IL-1β and TNF release from living cells under conditions where viruses would otherwise subvert macrophages for replication without triggering any cytokine production. We define the TRIM21/cGAS/NLRP3 axis as a primary driver of the inflammatory response to AdV in primary human macrophages.

# Results

## Monoclonal and polyclonal anti-AdV antibodies trigger inflammasome activation in human macrophages

Antibody-driven enhancement of inflammatory responses to AdV has previously been studied using non-specific pooled human serum as a source of anti-AdV antibodies (Muruve *et al*, 2008; Zaiss *et al*, 2009; Barlan *et al*, 2011a; Eichholz *et al*, 2016). Multiple pathways and mechanisms have been proposed in these studies. To dissect the role of antibodies in modulating human macrophage responses to AdV infection further, we utilised a monoclonal mouse-human chimeric IgG1 antibody against the hexon capsid of human Ad5 (h9C12) and compared it to pooled serum IgG (IVIg) in enhancing inflammatory responses to an Ad5-GFP vector (hereafter AdV). We measured the release of IL-1β to quantify inflammasome activation and the secretion of TNF, as anti-AdV antibodies had previously been shown to potently induce its transcription in mouse fibroblasts (Watkinson *et al*, 2015). AdV alone (50,000 pp/cell) barely elicited mature IL-1β and TNF release from differentiated THP-1 cells, despite infecting ~95% of cells, highlighting its ability to infect undetected (Fig 1A and B). However, in the presence of either IVIg or h9C12, both cytokines were detected in cell supernatants in a time-dependent manner (Fig 1B). We confirmed that we were detecting cleaved IL-1β in the supernatants by Western blot (Fig 1C). While AdV alone seemed to enhance expression of pro-IL-1β protein in PMA differentiated THP-1s, cleaved IL-1β was only detected in the presence of antibody. We repeated these experiments in primary human monocyte-derived macrophages (HMDM) and observed a similar lack of inflammasome activation when challenging with virus alone but a robust response in the presence of antibody (Fig 1D). Addition of antibody alone did not trigger a response. Importantly, IL-1β production was only seen in HMDM primed with the TLR3 agonist poly I:C (pI:C), consistent with two-step inflammasome activation and in contrast to the response in THP-1s, where PMA differentiation is also sufficient as a priming stimulus to drive expression of pro-IL-1β (Dostert *et al*, 2008) and where AdV alone enhances pro-IL-1β expression (Fig 1C). Finally, while priming in HMDM was obligatory for antibody-induced IL-1β secretion, TNF secretion was also induced by antibody–virion complexes even in un-primed HMDM (Fig 1D).

### AdV–antibody complex-dependent IL-1β and TNF release can occur without concurrent cell death

To further demonstrate the importance of antibodies in promoting viral detection and cytokine secretion, we undertook a titration of virus or virus:antibody complex (i.e. number of virus particles per cell). Both h9C12 and IVIg enhanced IL-1β and TNF release from THP-1 cells across a range of doses of virus–antibody complex, although IVIg caused more THP-1 death as measured by LDH release (Fig 2A and E). Importantly, neither cytokine was produced by cells infected with virus alone. Using the highest dose of virus (50,000 pp/cell), we then measured whether the concentration of antibody per virus impacted cytokine production in THP-1s. Indeed, we found that cytokine release was proportional to the amount of antibody per virus (Fig 2B). We also noted that none of the h9C12 antibody concentrations substantially increased THP-1 cell death compared to virus alone, while the two high doses of IVIg (20 and 4 mg/ml) increased THP-1 cell death. The only dose of AdV-IVIg (0.8 mg/ml) that did not trigger cell death in THP-1s did not trigger any cytokine release in these cells (Fig 2B). We performed the same titrations in HMDM and confirmed that, as in THP-1, h9C12 triggers cytokine release without concomitant cell death (Fig 2C). We also noted that in HMDM the dose of 0.8 mg/ml IVIg triggered cytokine release without the strong increase in cell death seen at the higher IVIg doses, suggesting that in HMDM the separation of cell death and cytokine release with IVIg is also possible (Fig 2D). We confirmed these cell death responses in THP-1s and HMDM by measuring LDH release and by measuring cell viability with a PrestoBlue assay (Fig 2E–H). We saw that only high-dose IVIG induced significant cell death in both cell types, and that this occurred independently of pI:C priming (Fig 2G and H). Taken together, this suggests that antibody opsonisation of AdV can trigger IL-1β or TNF release under conditions that do not trigger concurrent cell death.

### h9C12-dependent enhancement of inflammasome responses to AdV is independent of increased uptake or lysosomal damage

Uptake of opsonised pathogens is a key function of macrophages, so we investigated whether antibodies were enhancing inflammasome activation by delivering more virus (and therefore more viral PAMPs) into the cell. To test this, we quantified uptake of viral genomes in HMDMs 2 h post-challenge with AdV. We found that h9C12 and 0.8 μg/ml IVIg had no effect on viral entry, while high-dose pooled IVIg (20 mg/ml, 4 mg/ml) dramatically increased uptake (Fig 3A). We also determined the effect of these antibodies on viral infection. To measure infection in HMDM, we used the same viral dose (50,000 pp/cell) but a combination of Ad5-GFP and Ad5-mCherry viruses in a 200:1 ratio, so that we could determine the percentage of infected cells using the mCherry signal by flow cytometry. Both h9C12 and IVIg neutralised virus infection (as determined by mCherry fluorescence) at a high dose of antibody (Fig 3B). Importantly, the increased viral uptake seen with IVIg in Fig 3A did not increase viral infection (Fig 3B), suggesting that this mechanism of cell entry is nonproductive for the virus. This is in contrast to the antibody-dependent enhancement (ADE) of infection that is seen for viruses such as Dengue (Lu *et al*, 2017). This is also in agreement with previous data showing an FcγR-dependent increase in the number of viral particles in THP-1s in the presence of serum (Zaiss *et al*, 2009). Previous

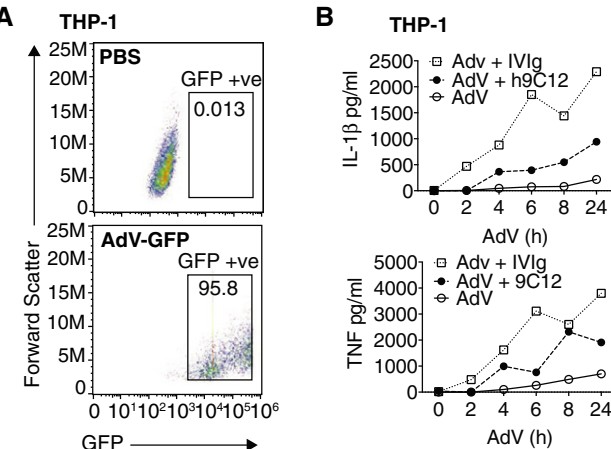

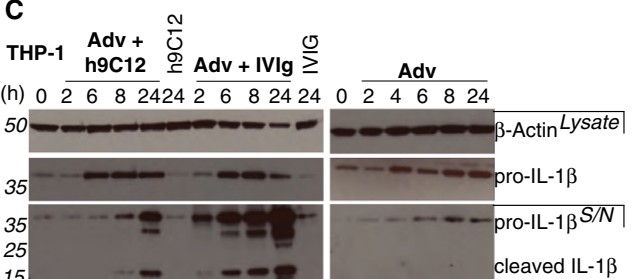

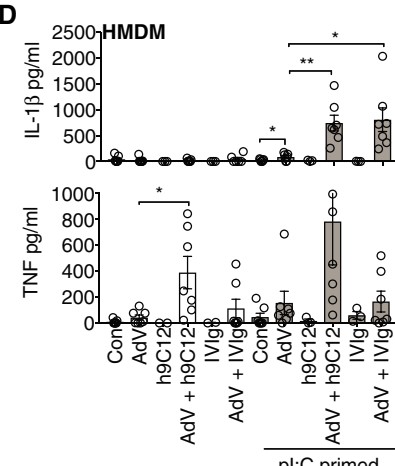

**Figure 1. Monoclonal and polyclonal anti-AdV antibodies trigger inflammasome activation in human macrophages.**

A   THP-1 cells were stimulated with (50,000 particles/cell) AdV-GFP and infection after 24 h measured by flow cytometry.

B,C   Cells were stimulated as in (A) with AdV-GFP or AdV-GFP complexed with 20 μg/ml h9C12 or 20 mg/ml human IVIg. Cell supernatants were harvested at indicated time-points, and cytokines measured by ELISA (B) or pro- and cleaved IL-1β were measured in whole cell lysates or in supernatants by Western blot (C). Data are representative of two independent experiments. (D) HMDM were primed with 10 μg/ml pI:C for 2 h and then stimulated for 16 h with virus +/− antibody. Cytokines in the supernatants were measured by ELISA (n = 7, mean ± s.e.m. *P ≤ 0.05, **P ≤ 0.005, paired, two-tailed t-test).

Source data are available online for this figure.

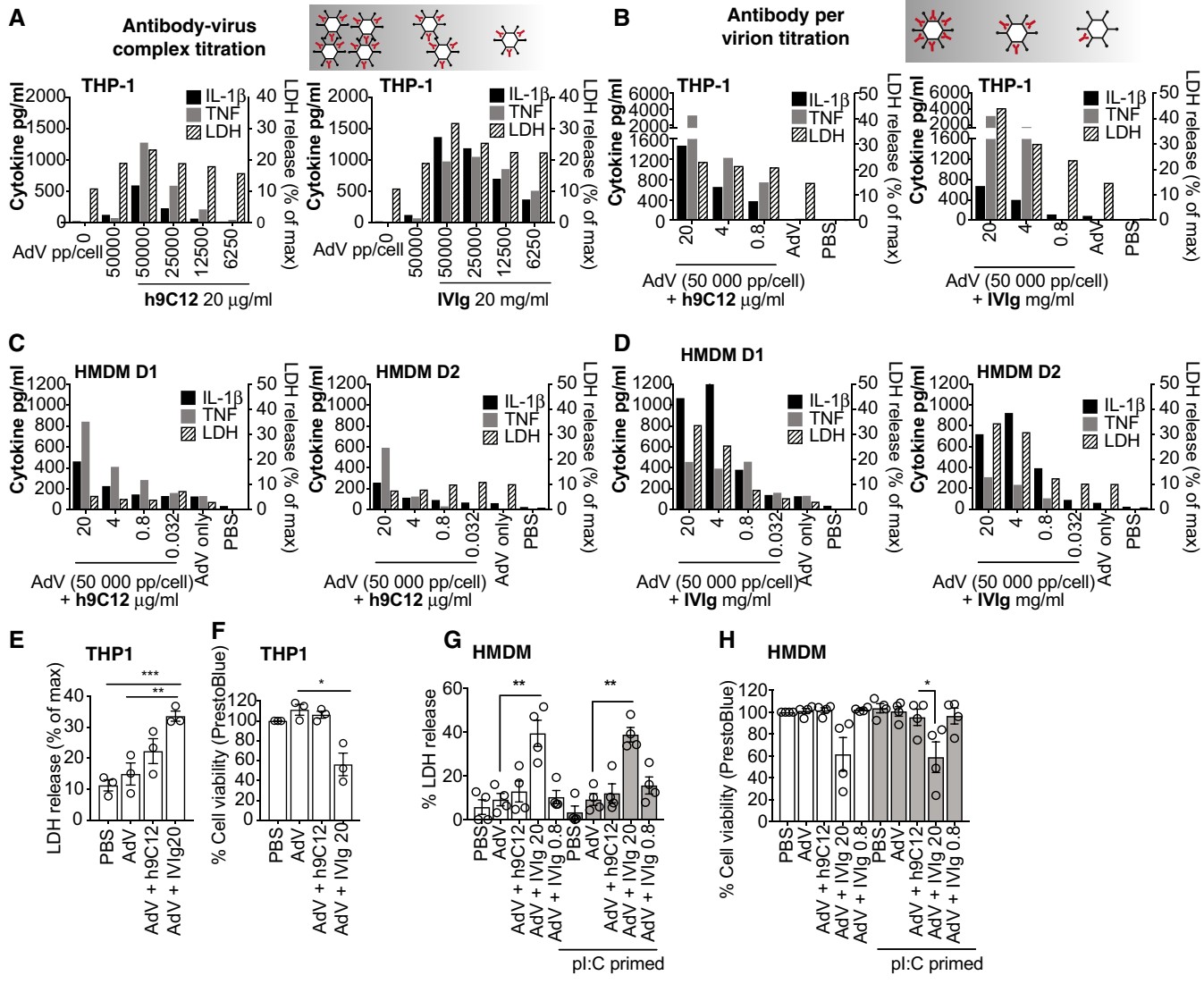

**Figure 2. Opsonised AdV-dependent IL-1β and TNF release can occur without concurrent cell death.**

IL-1 β, TNF (left axis) and LDH release (right axis) in the cell supernatants were measured 16 h after stimulation.

A      AdV-GFP was incubated with 20 μg/ml h9C12 antibody or 20 mg/ml IVIg in PBS. These AdV-Ab complexes were then diluted 1:2 to give final doses of viral particles being added to WT THP-1 cells (with antibody complexed) as indicated. A single experiment representative of three independent experiments is shown.

B      THP-1s were stimulated with a constant dose of AdV-GFP (50,000 pp/cell) and complexed with decreasing antibody concentrations as indicated. A single experiment representative of two independent experiments is shown.

C, D   HMDM were stimulated directly (TNF) or primed (IL-1β, LDH) with AdV-Ab complexes as in (B). Data for each individual donor (two total) are shown.

E, F   THP-1s were stimulated with AdV (50,000 pp/cell) complexed with h9C12 (20 μg/ml) or IVIg (20 or 0.8 mg/ml) for 16 h, and cell death was measured by LDH release (E) or cell viability measured by PrestoBlue assay (F) ($n = 3$, mean ± s.e.m. *$P \leq 0.05$, **$P \leq 0.005$, ***$P \leq 0.001$ unpaired, two-tailed $t$-test).

G, H   HMDM were primed, or not, and stimulated with AdV (50,000 pp/cell) complexed with h9C12 (20 μg/ml) or IVIg (20 or 0.8 mg/ml) for 16 h, and cell death was measured by LDH release (G) or cell viability measured by PrestoBlue assay (H) ($n = 4$, mean ± s.e.m. *$P \leq 0.05$, **$P \leq 0.005$, paired, two-tailed $t$-test).

studies in THP-1s have also shown that antibody-containing serum routes AdV into the lysosome (Barlan *et al*, 2011a). As adenoviruses have membrane lytic capability (Greber, 2016), lysosomal damage resulting from accumulated opsonised AdV has been proposed to trigger NLRP3 activation and IL-1β release in THP-1s (Barlan *et al*, 2011a). We reasoned that the lysosome may be the destination during increased non-productive viral entry in HMDMs at high IVIg concentrations and that lysosomal damage may be the trigger for the robust IL-1β release we observed at high concentrations of IVIg. We therefore

assessed lysosomal damage in HMDMs by measuring acridine orange (AO) fluorescence, which fluoresces red at low pH (i.e. upon accumulation in lysosomes), and fluoresces green at higher pH (i.e. in the cytosol). A loss of red signal is therefore indicative of lysosomal damage (Duewell & Latz, 2013). High-dose pooled IgG triggered loss of AO red fluorescence in HMDM 4 h post-infection, which no longer occurred when the concentration of IgG was reduced (Fig 3C). This is in agreement with the finding that lysosomal cathepsin B release in THP-1s is promoted by Ad5 + serum (Barlan *et al*, 2011a). We used

the fixable cell viability dye eFluor 780 (e780) to determine whether extensive lysosomal damage is compatible with cell death. Even 4 h post-infection, AdV-IVIg at 20 mg/ml caused approximately 20% of cells to die as determined by being positive for the e780 dye (Fig 3D). When we analysed the mean fluorescence intensity (MFI) of the AO red fluorescence in live cells versus dead cells for the AdV-IVIg condition, we found that the e780-positive cells had low AO red fluorescence, suggesting they had undergone extensive lysosomal damage (Fig 3E). Taken together, these results are consistent with the previously proposed model in which high-dose IVIg causes increased nonproductive virion uptake via FcγR's, trafficking to the lysosome and subsequent lysosomal damage that causes cytotoxicity and IL-1β release. However, IL-1β release was also observed at more modest levels of IVIg and when using h9C12—despite the fact that under these conditions there was no significant lysosomal damage, increased virion uptake or cytotoxicity. This suggests that there must be other mechanisms, whereby antibodies promote inflammasome activation.

### IL-1β and TNF release in response to AdV and h9C12 monoclonal antibody is TRIM21 dependent

To investigate how antibodies are provoking IL-1β release in the absence of lysosomal damage and cell death, we mutagenised the Fc region on h9C12 to selectively ablate interactions with different Fc receptors. We utilised a h9C12-H433A mutant that has previously been shown to prevent interaction with the cytosolic Fc receptor TRIM21 and a h9C12-LALA (L234A/L235A) mutant that inhibits interaction with FcγRs (McEwan *et al*, 2012) (Fig 4A). Testing these variants during AdV infection of HMDMs revealed that the Fc receptor binding deficient mutant h9C12-LALA was still able to induce IL-1β release, indicating that surface Fc receptors are unlikely to be playing a role in inflammasome activation in this context (Fig 4B). In contrast, h9C12-H433A gave a significantly reduced IL-1β response demonstrating that TRIM21 is required for IL-1β release in response to AdV and antibody immune complexes. TRIM21 has previously been shown to trigger immune signalling by virtue of its E3 ubiquitin ligase activity (McEwan *et al*, 2013). In response to opsonised AdV, TRIM21 mediates pro-inflammatory cytokine expression and this is prevented by h9C12-H433A mutation (McEwan *et al*, 2013). Consistent with this data, we found that h9C12-H433A but not h9C12-LALA abolished TNF secretion in response to AdV and antibodies in un-primed HMDMs (Fig 4C). These results suggest that TRIM21 is responsible for a potent inflammatory response in primary human macrophages. To provide further support that TRIM21 is involved in inflammasome activation, we complemented our antibody mutant approach by creating a Trim21-deficient THP-1 cell line using a lenti-CRISPR/Cas9 system (Fig 4D). IL-1β release in response to AdV and h9C12 was completely ablated in the Trim21-deficient THP-1s, while AdV-IVIg (20 mg/ml)- induced IL-1β was reduced. Trim21 deficiency did not affect cytokine release in response to transfected double-stranded DNA (HT-DNA) or to the NLRP3 agonist Nigericin (Fig 4E). The TNF response to both AdV-h9C12 and AdV-IVIg was also completely ablated in the TRIM21-deficient THP-1s (Fig 4F), though a slight defect in HT-DNA-induced TNF responses was also observed, which could indicate further roles for TRIM21 in regulating pro-inflammatory responses. We could not investigate whether TRIM21 deficiency would inhibit low-dose IVIg-induced cytokine release (as in

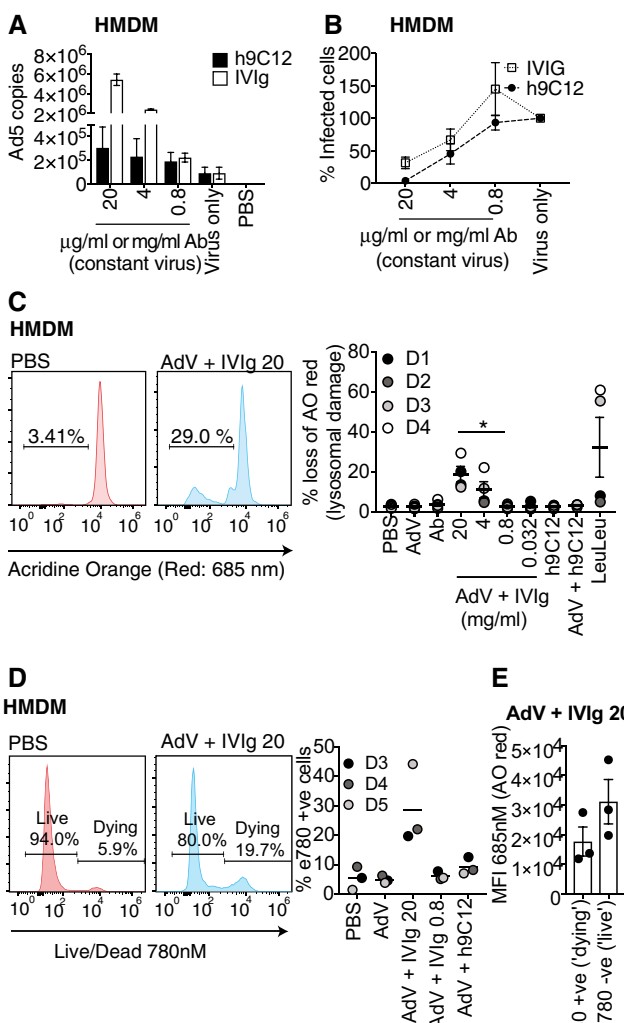

**Figure 3. h9C12-dependent enhancement of inflammasome responses to AdV is independent of increased uptake or lysosomal damage.**

A   HMDM were stimulated with AdV (50,000 pp/cell) complexed with antibody at indicated doses for 2 h, and viral genomes were measured by qPCR (*n* = 4, mean ± s.e.m).

B   AdV-mCherry was added to AdV-GFP at a ratio of 1:200, and 50,000 pp/cell of this mix was complexed with h9C12 and IVIg and indicated doses. HMDM were stimulated with these complexes for 16 h and infection measured as mCherry fluorescence by flow cytometry. In virus alone the percentage of mCherry-positive cells was ~20%, the relative infection with antibodies was normalised to virus alone as 100% (average + s.d. of two independent donors).

C   HMDM were stimulated for 3 h with AdV-Ab complexes as in (A, B) or with LeuLeu at 10 μM, and lysosomal damage measured by loss of acridine orange red fluorescence using flow cytometry. Representative plots showing gating are shown on left (*n* = 4, mean ± s.e.m. *\*P* ≤ 0.05, paired, two-tailed *t*-test).

D   HMDM were stimulated as in (C), and cell death was assessed by an increase in e780 fluorescence. The mean of three independent donors is shown.

E   The MFI of the AO red channel (685 nM) for the populations determined as e780 positive (dying) or e780 negative (live) for the condition of AdV + IVIg 20 was measured and plotted as mean + SEM of the three donors shown in (D).

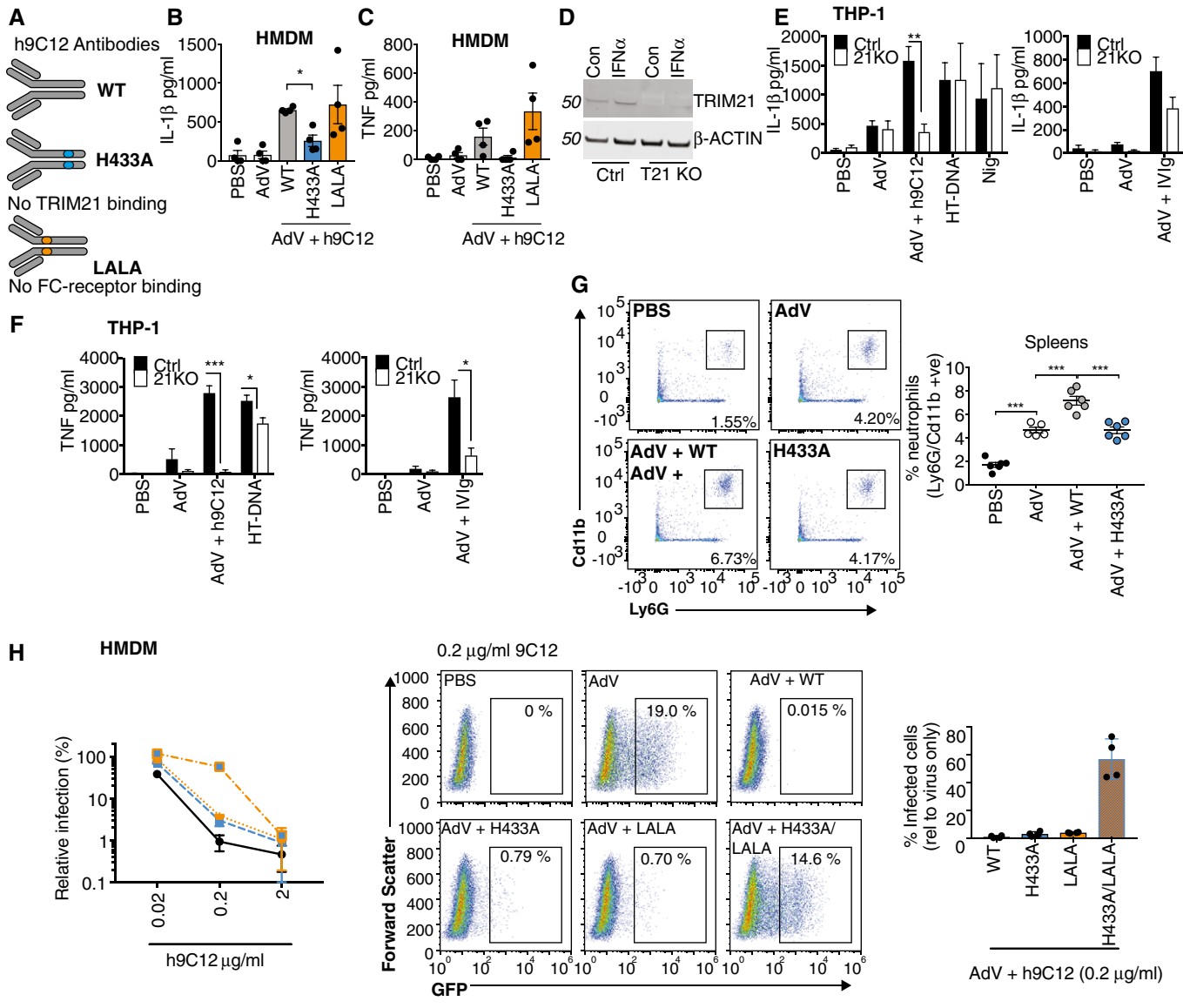

**Figure 4. IL-1β and TNF release in response to AdV and h9C12 monoclonal antibody is TRIM21 dependent.**

A    Schematic showing the h9C12 antibody mutants that no longer engage Trim21 (H433A) or FC receptors (LLAA).

B, C    (B) Primed or (C) un-primed HMDM were stimulated with AdV (50,000 pp/cell) +/− h9C12 (20 μg/ml) for 16 h (n = 4, mean ± s.e.m. *P ≤ 0.05, paired, two-tailed t-test).

D    Ctrl or Trim21-deficient THP-1s were stimulated for 4 h with 1,000 U/ml IFNα and TRIM21 expression measured by immunoblot.

E, F    WT or TRIM21-deficient THP-1s were stimulated with AdV (50,000 pp/cell) and 20 μg/ml h9C12 or 20 mg/ml IVIg or with 200 ng/well HT-DNA or 10 μM Nigericin for 16 h, and IL-1β (E) and TNF (F) measured in the supernatant by ELISA (n = 4 (E) or n = 3 (F), mean ± s.e.m. *P ≤ 0.05, **P ≤ 0.005, ***P ≤ 0.001 unpaired, two-tailed t-test).

G    WT mice were injected i.v. with 2.5 μg WT or H433A h9C12 antibody and then the next day injected i.v. with 2.5 × 10^11 pp AdV-GFP. 4 h later spleens were harvested and neutrophil influx measured by flow cytometry (n = 6 mean ± s.e.m. ***P ≤ 0.001 unpaired, two-tailed t-test).

H    HMDM were stimulated with AdV-GFP (250 pp/cell) in the presence of h9C12 antibodies at indicated doses. Infection was measured after 24 h by flow cytometry. Infection relative to AdV alone is shown graphed on the left (n = 4 mean ± s.e.m.). Representative plots and a comparison of the different h9C12 mutants at 0.2 μg/ml are also shown in the graph on the right (n = 4, mean ± s.e.m.).

Source data are available online for this figure.

the HMDM), as this dose did not trigger IL-1β or TNF release in THP-1s (Fig 2B).

Inflammasome-derived IL-1β is a major driver of neutrophilia *in vivo* and acts by promoting the secretion of neutrophil-recruiting chemokines such as MIP-2 and KC (Chen & Schroder, 2013). To assess whether AdV and antibody trigger neutrophil recruitment *in vivo* in a TRIM21-dependent manner, we i.v. infected WT mice with virus alone, virus and WT h9C12 or virus and H433A h9C12 and assessed splenic neutrophil recruitment by flow cytometry. While AdV alone increased the total number of neutrophils in the

spleen, this was significantly increased in the presence of WT h9C12, but not H433A h9C12, (Fig 4G). This indicates that TRIM21 plays a role in neutrophil recruitment and hence the inflammasome response during *in vivo* infection.

The interception of incoming antibody-coated AdV during macrophage infection suggested that in addition to stimulating the inflammasome TRIM21 might operate to protect these cells from becoming a niche for viral replication. To investigate this, we determined the efficiency of productive macrophage infection in the presence of antibody by measuring the synthesis of a virally encoded reporter gene (GFP). AdV infection decreased with increasing h9C12 antibody concentration, indicating dose-dependent neutralisation (Fig 4H). Using H433A h9C12, which ablates TRIM21 binding, did not prevent neutralisation. This is in contrast to non-immune cells such as 293Ts where h9C12 neutralisation of AdV is completely TRIM21 dependent (Bottermann *et al*, 2016). Using a LALA h9C12 with reduced FcγR binding also failed to prevent neutralisation. However, when H433A was combined with the LALA mutations to inhibit binding to both Trim21 and the FcγRs, neutralisation was prevented at intermediate antibody concentrations (Fig 4H). This suggests that there are multiple redundant routes of viral entry, in which virus: antibody complexes can engage with either FcγRs or TRIM21. This is consistent with a requirement for multiple inflammasome activation mechanisms to detect infection.

## TRIM21 is involved in inflammasome activation rather than enhancing pro-IL-1β expression

TRIM21 has previously been shown to activate NFκB and IRF3 signalling pathways (McEwan *et al*, 2013). NFκB activation is required for inflammasome priming, which may include post-translational modifications of NLRP3 as well as expression of pro-IL-1β. While AdV and antibody did not trigger IL-1β release in resting primary macrophages (Fig 1D), we wanted to assess whether TRIM21's potentiation of the inflammasome was due to enhanced expression of pro-IL-1β. AdV alone did not induce *Il1b, Tnf or Ifnb* mRNA expression at 3 h post-infection, but AdV in the presence of WT h9C12 induced their expression approximately 10-fold. This induction was completely TRIM21 dependent, as h9C12-H433A did not increase transcription (Fig 5A). However, the magnitude of increased pro-IL-1β transcription was considerably lower than following stimulation with pI:C at the dose used to prime HMDM for inflammasome activation, which increased IL-1β mRNA expression approximately 200-fold. Consistent with this, while stimulation with pI:C or the TLR4 agonist lipopolysaccharide (LPS) substantially induced pro-IL-1β protein expression, stimulation with virus and h9C12 did not (Fig 5B). Stimulation with AdV and IVIg also did not induce *Il1b, Tnf or Ifnb* mRNA expression significantly, nor was pro-IL1β protein significantly upregulated with IVIg (Fig 5A and B). The NLRP3 inflammasome is also regulated in part by NFκB-dependent transcription of NLRP3. However, we found that neither AdV-h9C12 nor AdV-IVIg induced NLRP3 mRNA (Fig 5C) or protein expression (Fig 5D). This indicates that the h9C12/TRIM21 pathway is unlikely to enhance IL-1β production via potentiating either pro-IL-1β or NLRP3 expression.

Previous studies have suggested that AdV primes the inflammasome by triggering TLR9 activation, leading to NFκB induction and upregulation of inflammasome components including pro-IL-1β

(Barlan *et al*, 2011b; Eichholz *et al*, 2016). However, we found that TLR9 was very lowly expressed in primary human macrophages (Fig 5E), consistent with reported gene expression profile data available in immGen (www.immgen.org), which may explain why there is poor priming with AdV and antibody in these cells. Priming and inflammasome activation can be differentiated mechanistically by measuring assembly rather than component expression levels. The adaptor ASC is a canonical inflammasome cofactor that undergoes higher-order assembly into large cellular structures known as "specks". The formation of these specks requires activation but precedes caspase-1 recruitment and IL-1β cleavage and has been widely used as a marker for the second step in inflammasome activity (Franklin *et al*, 2014). To assess whether TRIM21 stimulates ASC specking, we used THP-1s stably expressing fluorescent ASC, differentiated them with PMA, then challenged with virus and antibody. We observed a significant increase in ASC specking in the presence of AdV and h9C12-WT, but not AdV and h9C12-H433A, suggesting that TRIM21 is required for ASC assembly in response to virus and h9C12 antibody (Fig 5F). This supports the idea that TRIM21 integrally contributes to inflammasome activation in response to virus and antibody.

## NLRP3 is required for inflammasome activation downstream of TRIM21

To determine how TRIM21 might be involved in inflammasome activation, we first investigated which other inflammasome sensors may be required. In HMDM, IL-1β release induced by AdV and h9C12 or IVIg was ablated by the specific NLRP3 inhibitor MCC950 (Coll *et al*, 2015), suggesting that NLRP3 is required for both TRIM21-dependent and TRIM21-independent, antibody-induced IL-1β release (Fig 6A). As expected, TNF release was not inhibited by MCC950 (Fig 6B). AdV-IVIg-induced cell death was not inhibited by MCC950, suggesting it is not pyroptosis (Fig 6C). Consistent with this result, NLRP3-, ASC- and caspase-1-deficient THP-1s no longer released IL-1β in response to AdV and antibody (Fig 6D). Furthermore, inhibition of caspase-1 by the inhibitor VX-765 inhibited IL-1β release in HMDM but not AdV-IVIg-induced cell death (Fig 6E), consistent with effects seen with MCC950. The precise molecular mechanism of NLRP3 activation remains unclear, but efflux of intracellular $K^+$ is required for activation of NLRP3 by many but not all characterised NLRP3 agonists (Gaidt & Hornung, 2018). To investigate whether this is also required during the response to AdV and antibody, we dose-dependently inhibited $K^+$ efflux with high extracellular KCl. The induction of IL-1β by AdV and antibody was blocked by high extracellular KCl, as were Nigericin and dsDNA induced IL-1β (Fig 6F). Again, TNF release was unaffected by high extracellular KCl (Fig 6F). This suggests that K+ efflux is required for NLRP3 activation in response to virus and antibody, rather than other potential mechanisms such as mitochondrial ROS generation. While the sensing of lysosomal damage as a result of increased viral uptake most likely explains IVIg-induced NLRP3 activation, how h9C12 triggers NLRP3 activation through TRIM21 is unclear. As we had observed TRIM21-dependent ASC specking in response to AdV-h9C12, we investigated the localisation of ASC and intracellular antibody-coated virus post-infection. We hypothesised that TRIM21 might form an anchor point for NLRP3 oligomerisation and ASC

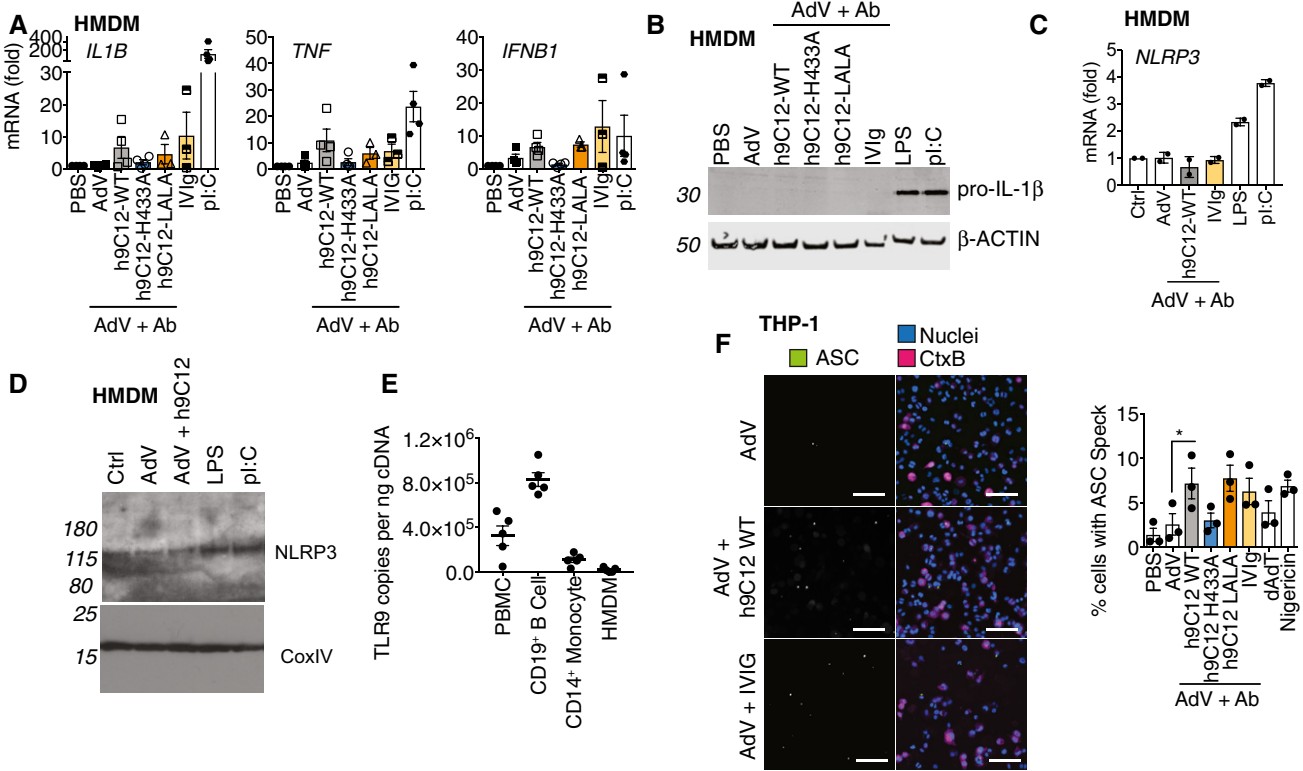

**Figure 5. TRIM21 is involved in inflammasome activation rather than enhancing pro-IL-1β expression.**

A   HMDM were stimulated with AdV (50,000 pp/cell) and antibody (h9C12; 20 μg/ml, IVIg: 20 mg/ml) or 10 μg/ml pI:C for 3 h and gene expression was measured by qPCR ($n = 4$ mean $\pm$ s.e.m).

B   HMDM were stimulated as in (A), or with 10 ng/ml LPS for 6 h and pro-IL-1β levels in the cytosol measured by Western blot. Blot is representative of three independent donors.

C   HMDM were stimulated as in (A), or with 10 ng/ml LPS for 3 h and NLRP3 mRNA expression measured by qPCR. Data show average $\pm$ s.d. of two independent donors.

D   HMDM were stimulated as in (B) and NLRP3 levels in the cytosol measured by Western blot. Blot is representative of two independent donors.

E   TLR9 mRNA levels from PBMCs, CD19[+ve] B cells, CD14[+ve] monocytes or HMDM derived from CD14[+ve] monocytes were assessed by qPCR, and copy number was determined relative to actin copy number ($n = 5$ mean $\pm$ s.e.m).

F   THP-1s expressing ASC-GFP were stimulated for 6 h with virus and antibody or 200 ng/well HT-DNA or 10 μM Nigericin, in the presence of the pan-caspase inhibitor zVAD-fmk. A representative image (scale bar 100 μm) and quantification of number of cells with ASC specks from three independent experiments (mean $\pm$ s.e.m, *$P \leq 0.05$, paired, two-tailed $t$-test) are shown.

Source data are available online for this figure.

specking. However, we saw no co-localisation between ASC and h9C12-coated virions at 3 h post-infection (Fig 6G). TRIM21 and NLRP3 are therefore both required for inflammasome activation in response to virus and h9C12 antibody but we observed no direct interaction between them.

**Trim21 exposes AdV genomes to cGAS and STING to trigger NLRP3-dependent inflammasome activation**

Previously, we have shown that TRIM21 potentiates sensing of incoming virions by the cytoplasmic nucleic acid sensors RIG-I and cGAS by exposing viral genomes upon catastrophic uncoating of capsids (Watkinson *et al*, 2015). This results in activation of IRF3 and NFκB and the induction of TNF transcription in fibroblasts. Nucleic acid sensing has been implicated in inflammasome activation, with one study describing a role for the DNA sensor AIM2 in AdV and antibody-mediated pyroptosis in human dendritic cells,

though notably, AIM2 was dispensable for IL-1β release (Eichholz *et al*, 2016). However, we did not detect AIM2 mRNA in our primary human macrophages (Fig 7A), consistent with gene expression profile data in immGen (www.immgen.org). A recent study from Gaidt and colleagues demonstrated that in human myeloid cells, AIM2 does not mediate the inflammasome response to cytosolic dsDNA but that this requires cGAS/STING signalling to drive NLRP3 activation (Gaidt *et al*, 2017). We therefore investigated whether TRIM21 may be potentiating this cGAS/STING-induced NLRP3 inflammasome in a similar way to cGAS-dependent TNF induction: by revealing the AdV genome. To confirm that the AdV capsid is being degraded in our experiments, we blotted for the major capsid-protein hexon in newly infected primary human macrophages. In the presence of h9C12, we observed efficient hexon degradation, consistent with TRIM21 targeting antibody-bound virus for proteasomal degradation as has been previously shown (Mallery *et al*, 2010; Hauler *et al*, 2012; Fig 7B). To directly determine

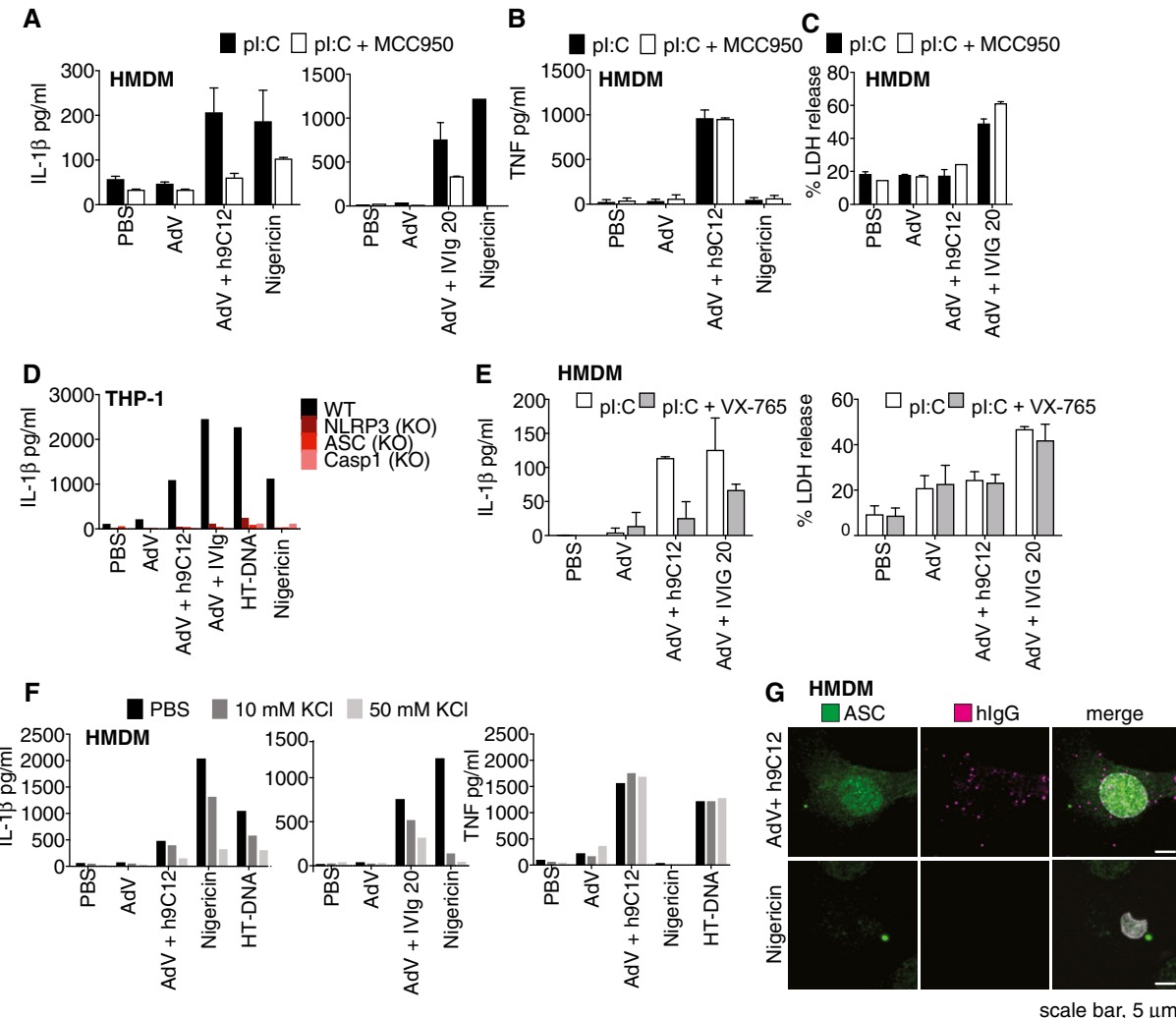

**Figure 6. NLRP3 is required for inflammasome activation downstream of TRIM21.**

A–C  Primed HMDM were treated with MCC950 (1 μM) for 30 min before stimulation overnight with AdV (50,000 pp/cell) and h9C12 (20 μg/ml) or IVIg (20 mg/ml) or 10 μM Nigericin. IL-1β (A) or TNF (B) or LDH release (C) was measured in cell supernatants. Data show average ± s.d. of two independent donors and are representative of three independent experiments.

D  WT, NLRP3, ASC or caspase-1-deficient THP-1s were stimulated as in (A) and IL-1β in the supernatant measured by ELISA. Data are representative of two independent experiments.

E  HMDM were primed with 10 μg/ml pI:C for 2 h, treated with 100 μM VX-765 for 30 min before stimulation for 16 h as in (A). IL-1β or LDH release was measured in cell supernatants. Data show average ± s.d. of two independent donors.

F  Primed HMDM were treated with KCl as indicated for 1 h, before being stimulated for a further 6 h as in (A). IL-1β and TNF were measured in cell supernatants by ELISA. Data are representative of three independent donors.

G  Primed HMDM were stimulated with AdV and 20 μg/ml h9C12 or 10 μM Nigericin for 3 h and then immunostained for ASC or intracellular antibody. Co-localisation was assessed by confocal microscopy. Data are representative of two independent donors.

whether cGAS and STING are involved in the inflammasome response to virus and antibody, we generated CRISPR knockout THP-1s (Fig 7C). We observed that IL-1β and TNF release in response to AdV and h9C12 was reduced in cGAS and STING-deficient THP-1s (Fig 7D). As expected, HT-DNA responses were impaired, and Nigericin and LPS responses in these knockout THP-1s were unaffected. We then utilised the newly described STING inhibitor H151 (Haag *et al*, 2018) to further investigate this pathway. We found that H151 very efficiently inhibited HT-DNA and cGAMP-induced IL-1β and TNF responses in WT THP-1s without

impacting Nigericin-induced IL-1β, showing that it specifically inhibited the STING response (Fig 7E). AdV-h9C12-induced IL-1β was inhibited by H151, suggesting that cGAS and STING are indeed involved in this pathway. However for AdV-IVIg, IL-1β release was less inhibited, consistent with a role for early lysosomal damage directly triggering NLRP3 activation. Unexpectedly, AdV-h9C12 and AdV-IVIg-induced TNF responses were not inhibited by H151 (Fig 7E). Finally, H151 also inhibited AdV-h9C12-induced ASC specking in THP-1s expressing ASC-GFP (Fig 7F), indicating that STING is upstream of NLRP3 activation.

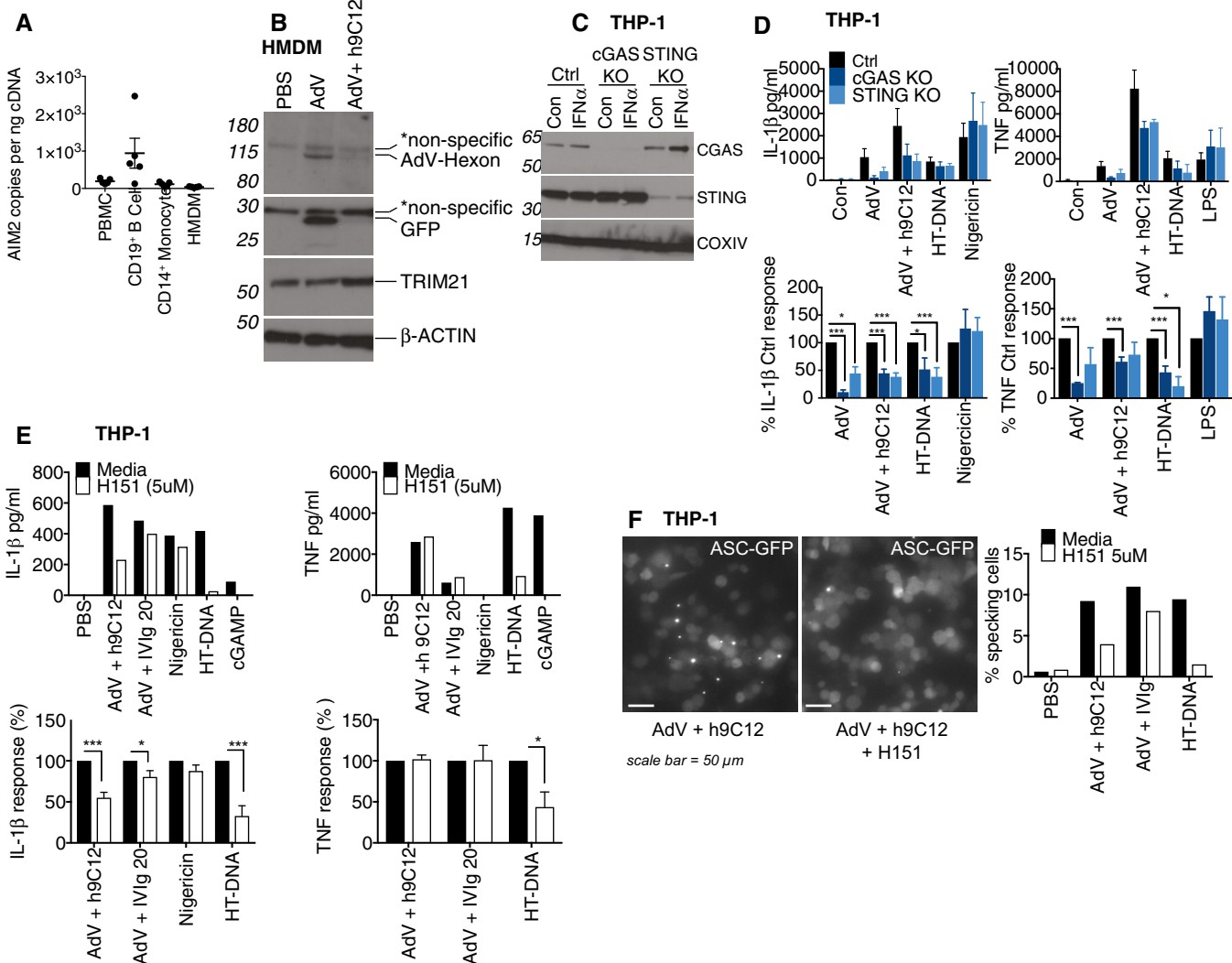

**Figure 7. Trim21 exposes AdV genomes to cGAS and STING to trigger NLRP3-dependent inflammasome activation.**

A  AIM2 mRNA levels from PBMCs, CD19[+ve] B cells, CD14[+ve] monocytes or HMDM derived from CD14[+ve] monocytes were assessed by qPCR, and copy number was determined relative to actin copy number (*n* = 5 mean ± s.e.m).

B  HMDM were stimulated with AdV (50,000 pp/cell) and 20 μg/ml h9C12 for 1 h, washed 2× with SFM and then whole cell lysates harvested after a further 5 h. Viral hexon and transgene (GFP) expression in the cytosol was assessed by Western blot. Data are representative of two independent experiments.

C  THP-1s expressing either a control guide RNA or targeting cGAS and STING were generated and stimulated with 1,000 U/ml IFN-α for 4 h and protein levels assessed by Western blot.

D  THP-1s deficient in cGAS and STING were stimulated with AdV (50,000 pp/cell) and 20 μg/ml h9C12, 200 ng/well HT-DNA, 10 μM Nigericin or 10 ng/ml LPS for 16 h. Data show combined data (mean ± s.e.m) of three experiments with absolute protein values (upper panel) or as % cytokine output of KO cells relative to Ctrl-treated cells (lower panel), *$P \leq 0.05$, ***$P \leq 0.001$ unpaired, two-tailed *t*-test).

E  WT THP-1s were treated with 5 μM H151 for 30 min before stimulation as in (D). Data in upper panel are representative of three independent experiments. Data in lower panel show combined data of these three experiments (mean ± s.e.m) showing H151-treated cells relative to media treated cells (*$P \leq 0.05$, ***$P \leq 0.001$ unpaired, two-tailed *t*-test).

F  ASC-GFP THP-1s were treated with 5 μM H151 for 30 min before stimulation with AdV-mCherry (50,000 pp/cell) + 20 μg/ml h9C12 or 20 mg/ml IVIg or 200 ng/well HT-DNA for 8 h. A representative image (scale bar 50 μm) and quantification of number of cells with ASC specks from one representative experiment of three are shown.

Source data are available online for this figure.

## Discussion

Our data highlight two distinct mechanisms that allow anti-viral antibodies to activate the inflammasome during infection of primary human macrophages. In the first mechanism, high-dose antibody enhances capture of AdV in what is likely a classic phagocytosis response. This results in lysosomal damage, NLRP3 activation, cell death and the release of IL-1β. These results are consistent with previous reports of serum-enhanced FcγR-dependent viral uptake (Zaiss *et al*, 2009) and lysosomal trafficking and damage (Barlan

*et al*, 2011a). In the second mechanism, antibodies activate the inflammasome without causing any change in virion uptake, lysosomal damage or cell death. Instead, antibody-coated virions are detected by the cytosolic antibody receptor TRIM21, which triggers both TNF and IL-1β release. Mechanistically, we propose that TRIM21 promotes inflammasome activation via the newly described cGAS/STING-induced NLRP3 inflammasome (Gaidt *et al*, 2017). This is in contrast to DCs where AIM2 is required for AdV-IVIg-induced cell death, though not for IL-1β release (Eichholz *et al*, 2016). TRIM21 has previously been shown to recruit the proteasome to degrade incoming viral capsids in non-immune cells (Mallery *et al*, 2010), and we observed similar capsid-protein loss in HMDMs. Capsid degradation by TRIM21 reveals viral genomes to nucleic acid sensors cGAS and RIG-I and promotes cytokine transcription in fibroblasts (Watkinson *et al*, 2015). The data presented here suggest that a similar molecular mechanism operates in macrophages to promote efficient inflammasome activation. Concomitant with inducing pro-inflammatory signalling, antibody-dependent degradation of viral capsids also neutralises viral infection. While TRIM21 is necessary and sufficient for proteasomal degradation of cytosolic antibody-bound targets, be they viral or cellular in non-immune cells such as fibroblasts (Clift *et al*, 2017; Bottermann *et al*, 2018), in primary macrophages we saw that inhibiting either TRIM21 or FcγR interactions did not impact antibody-mediated neutralisation. However, when both pathways were inhibited, antibody-mediated neutralisation was compromised. This suggests that the TRIM21 and FcγR pathways function in a redundant manner to neutralise AdV, such that you only block neutralisation when both of these pathways are suppressed. Nevertheless, these results are consistent with our observation of two independent systems of inflammasome activation by antibody and are likely a consequence of the different routes of AdV entry. In the presence of antibody, AdV is taken up by FcγR and routed to the lysosome where it is rendered non-infectious. AdV that escapes FcγR uptake, such as in the case of the LALA mutant, enters the cytosol but is then detected by TRIM21, which causes catastrophic uncoating and neutralisation. The expression of both surface (FcγR) and cytosolic (TRIM21) antibody receptors in primary macrophages therefore provides not only sensors for inflammasome activation but also potent mechanisms to prevent the cells from being colonised.

TRIM21 is an E3 ubiquitin ligase that can synthesise unanchored K63 chains, which directly activate NFκB signalling pathways (McEwan *et al*, 2013). Consistent with this, we observed that TRIM21 stimulates the NFκB pathway to trigger TNF release in primary human macrophages. However, while this is accompanied by a significant increase in *Il1b*, *Tnf* and *Ifnb1* mRNA expression, it is not sufficient for TRIM21 increase pro-IL-1β to detectable levels. This is in contrast to results seen in primary human monocyte-derived DCs where AdV-IVIg immune complexes were also sufficient to prime cells for pro-IL-1β expression (Eichholz *et al*, 2016). In DCs, this was partially TLR9 dependent, and AdV-IVIg immune complexes triggered pyroptotic cell death. In contrast, we found in HMDMs that the cell death accompanying IL-1β release in response to AdV-IVIg was unlikely to be pyroptosis, as it was not inhibited by MCC950 or VX-765 (Fig 6C and E), and that for h9C12, IL-1β release occurred without significant cell death.

While TRIM21-mediated IL-1β release was reduced in cGAS and STING knockout THP-1s, it was not completely abolished as in

caspase-1, ASC and NLRP3 knockouts. Similarly, H151 inhibited IL-1β release induced by AdV-h9C12 to about 50%, while the inhibition by MCC950 was much more pronounced. It is therefore possible that TRIM21 can activate NLRP3 through both cGAS/STING-dependent and cGAS/STING-independent mechanisms. TRIM21 is prone to aggregation (in so called cytoplasmic bodies; Reymond *et al*, 2001) upon activation and whether this causes a change in cellular homeostasis that can be sensed by NLRP3 remains to be determined. TRIM21-induced NLRP3 activation requires K$^+$ efflux but this is also consistent with sensing of AdV genomes via cGAS and STING (Gaidt *et al*, 2017). It does not appear that TRIM21 recruited to an antibody-coated virion acts as a scaffold for inflammasome assembly as, although TRIM21 induced ASC specking in a cell line assay upon infection, there was no co-localisation of specks with virions in HMDMs. Additionally, while ASC specks were observed in HMDMs, there was not a complete relocalisation of diffuse cytosolic ASC to a single point within the cell. This is in contrast to canonical NLRP3 agonists such as Nigericin or ATP but similar to other inflammasome agonists, such as bacterial peptidoglycan that can trigger IL-1β release from living macrophages, without all the ASC in the cell localising to the one speck (Evavold *et al*, 2018). The dynamics of ASC speck formation, and indeed possibly ASC speck dissolution, are still incompletely understood, and further research into this area with pathogens rather than model ligands may be required.

The co-operation of various macrophage PRRs, such as TRIM21 with cGAS and NLRP3, to integrate signals derived from different host and pathogen molecules presents a way for immune cells to tailor the pro-inflammatory response to the nature of the incoming pathogen. Moreover, in the case of non-enveloped viruses, detecting host serum factors that have been mis-localised upon infection represents an effective way to detect constantly evolving pathogens, particularly pathogens which may antagonise inflammasomes (Maltez & Miao, 2016). Specifically, our data illustrate how adaptive immunity, in the form of antibodies, promotes innate immunity, in the form of the inflammasome. We show that this is crucial because pathogens like Ad5 can infect primary human macrophages without triggering an inflammasome response.

The importance of inflammasomes and IL-1β in particular in protecting against infection has been primarily studied in the context of bacterial infections. However, IL-1β and inflammasome signalling pathways are required for the clearance of Mouse adenovirus-1 (MAV1) infections *in vivo* (Castro-Jorge *et al*, 2017). Similarly, TRIM21 is required for efficient antibody protection against MAV1 infection *in vivo* (Vaysburd *et al*, 2013). Quantifying what part of the protective effect of antibodies during AdV infection is due to inflammasome activation, and signalling remains to be determined. Of the two mechanisms of antibody-induced inflammasome activation we describe here, we speculate that TRIM21 might be more important at the earlier stages of infection where antibody occupancy on each virion is low and most viruses enter the cell via a natural infectious route. This is suggested by the fact that the TRIM21 mechanism of IL-1β release is detectable at low antibody doses, while IL-1β release following redirection to the lysosome requires higher antibody occupancy, presumably for efficient FcγR cross-linking. This would agree with previous data showing that TRIM21 neutralises AdV at low antibody occupancy (McEwan *et al*, 2012). Antibody paratope may also influence the relative

importance of different antibody-induced inflammasome mechanisms, with a dominant anti-hexon repertoire (such as h9C12 used here) more likely to favour a TRIM21-based response. Understanding how antibodies enhance inflammasome responses may contribute to the ongoing engineering of adenoviral vectors for gene therapy, particularly as a recent study shows TRIM21 contributes significantly to their neutralisation (Bottermann *et al*, 2018). Inflammatory signalling is also inextricably linked to autoimmunity. A recent study showed that macrophages from lupus prone mice had defects in lysosomal maturation that allowed for escape of autoimmune complexes into the cytosol (Monteith *et al*, 2016). This included sensing of DNA–antibody complexes by TRIM21 and AIM2 to activate NFκB and inflammasome pathways, respectively. This raises the possibility that TRIM21 may coordinate with other macrophage PRRs in a pathogenic manner to drive autoimmune disease.

# Materials and Methods

## Cells

THP-1 cells were obtained frozen from ATCC and maintained in RPMI 1640 media supplemented with 10% FBS and sodium pyruvate. For differentiation, cells were seeded for assays in the presence of 200 nM phorbol-myristate-acetate and left to rest for 48 h. The media on the cells was changed to serum-free RPMI before further stimulation. THP-1 ASC-Cerulean cells were a gift from B. Franklin and were generated as described in Franklin *et al* (2014). CRISPR/Cas9 knockout single-guide RNA (sgRNA) against targets (Table 1) was incorporated into the lenti-CRISPR v2 plasmid (Addgene), and VSV-G pseudotyped lentiviral particles were generated by three-plasmid transfection of 293T with Fugene-6 (Promega), using 1 μg HIV-1 Gag-Pol expression plasmid and 1 μg VSV-G expression plasmid pMD2.G (GenScript), and 1.5 μg lenti-CRISPR v2. $5 \times 10^5$ THP-1s were transduced with 1 ml unconcentrated 293T viral supernatant in the presence of 8 μg/ml polybrene (Santa Cruz), and selected with 2.5 μg/ml puromycin. Polyclonal cell line loss of protein expression was confirmed by immunoblot.

Human peripheral blood mononuclear cells (PBMCs) were purified from buffy coats (ethical approval REC 16/LO/0997) from the National Health Service Blood and Transplant (Cat no: NC07 (Buffy Coats 50 ml)) by centrifugation over a Ficoll-Paque density gradient (GE Healthcare). CD14$^{+ve}$ monocytes were further purified by positive selection using CD14 microbeads (Miltenyi, Cat no. 130-050-201). Monocytes were plated in 6-well TC dishes with $2 \times 10^6$ cells

per well and differentiated over 7 days into macrophages in RPMI with 10% FCS and P/S in the presence of 40 ng/ml recombinant human MCSF (R & D systems, cat no: 216-MC-025). Macrophages were harvested by incubating cells in ice-cold PBS with 2% FCS and 5 mM EDTA for 10 min at 4°C, and harvesting cells with a cell scraper. Both THP-1 and HMDM cells were seeded at 250,000 cells per well of a 24-well plate, or 50,000 cells per well of a 96-well plate. Macrophages were primed as indicated in the figure legends for 3 h then washed twice with serum-free RPMI. Assays with antibody and virus were all done in serum-free medium. CD19$^{+ve}$ B cells were isolated using CD19 microbeads (Miltenyi Cat no. 130-050-301).

## Viruses, Antibodies and other reagents:

Human AdV type five vector (ΔE1, ΔE3) expressing GFP or mCherry (AdV) was purchased from ViraQuest. Pooled human serum IgG (IVIg) was purchased from Sanquin. Recombinant mouse-human chimeric monoclonal IgG1 h9C12 anti-hexon antibody and its mutant versions were generated as previously described (McEwan *et al*, 2012). AdV was incubated for 1 h at room temperature in PBS at a ratio of 1:1 with either 20 μg/ml h9C12 or 20 mg/ml IVIg and was added to a final concentration of 50,000 physical particles per cell. LPS (E.Coli 0111:B4, cat code: tlrl-eblps) and poly(I:C) (polyinosinic-polycytidylic acid high molecular weight, cat code: tlrl-pic) were purchased from InvivoGen. DNA from herring testes (HT-DNA) was purchased from Sigma (Catalogue no: D6898-250MG). Recombinant human IFN-α (Catalogue no: 300-02AA) was purchased from PeproTech. MCC950, formerly known as CRID3 (CAS no: 256373-96-3), was purchased from Tocris (Catalogue No: 5479). L-leucyl-L-leucine methyl ester (hydrochloride) (CAS no: 6491-83-4) was reconstituted to 1 mM in DMSO and diluted to 100 μM in RPMI media along with 200 μM HEPES buffer, and used at a final concentration of 10 μM. Nigericin sodium salt was purchased from Enzo Life Sciences (Catalogue no: BML-CA421-0005) and was reconstituted at 10 mM in ethanol. VX-765 (Catalogue no:inh-vx765i-1) and H151 (Catalogue no: inh-h151) were purchased from InvivoGen and reconstituted in DMSO at 25 mM and 35.8 mM, respectively. H151 was further diluted to 1 mM in DMSO before dilution to final concentration in serum-free media.

## ELISAs

ELISAs against human IL-1β (Catalogue no: DY201) and TNF (Catalogue no: DY210) were purchased from R & D systems and performed according to manufacturer's instructions.

## LDH assay

The lactate dehydrogenase cytotoxicity assay (Thermofisher, Catalogue no: 88953) was performed according to manufacturer's instructions.

## PrestoBlue cell viability assay

The PrestoBlue cell viability assay (Thermofisher, Catalogue no: A13261) was performed according to manufacturer's instructions.

**Table 1. CRISPR target sites used with PAM regions in bold.**

| Target | Guide RNA | References |
|--------|-----------|------------|
| cGAS | GAACTTTCCCGCCTTAGGCAG**GGG** | Wassermann *et al* (2015) |
| STING | ATCCATCCCGTGTCCCAG**GGG** | Wassermann *et al* (2015) |
| NLRP3 | GCTAATGATCGACTTCAATG**GGG** | Schmid-Burgk *et al* (2015) |
| ASC | GCTGGAGAACCTGACCGCCG**AGG** | Schmid-Burgk *et al* (2015) |
| Caspase 1 | ATTGACTCCGTTATTCCGAA**AGG** | Schmid-Burgk *et al* (2015) |
| TRIM21 | AGAAAGCGCTGCCGGCACAC**AGG** | This paper |
| Ctrl (GFP) | CGTCGCCGTCCAGCTCGACC**AGG** | This paper |

## Immunoblots

Cells were plated at 250,000 cells per well of a 24-well plate and stimulated as indicated in serum-free medium. 500 μl of supernatants was collected and precipitated using methanol/chloroform extraction as described in Jakobs *et al* (2013). Briefly, 500 μl MeOH and 125 μl chloroform were added to 500 μl supernatant, vortexed briefly and spun at 13,000 *g* for 5 min. The upper layer was removed and a further 500 μl MeOH added to each sample, vortexed and spun at 13,000 *g* for 5 min. The supernatant was completely aspirated, and the protein pellet was resuspended directly in NuPAGE LDS Sample buffer with 100 μM DTT and heated at 95°C for 10 min. Cells lysed in RIPA buffer (CST-9806) supplemented with a protease inhibitor cocktail (Roche), spun at 14,000 *g* for 10 min and cleared lysates mixed with NuPAGE LDS Sample Buffer and heated at 95°C for 10 min. Samples were run on NuPAGE 4–12% Bis-Tris gels (Thermo Fisher) and transferred onto nitrocellulose membrane. Antibody incubations were performed in PBS with 5% milk and 0.1% Tween-20. Primary antibodies were as follows: goat anti-IL-1β (BAF201, R & D Systems, 1:500), mouse anti-TRIM21 ((D-12, Santa Cruz Biotechnology sc-25351; 1:500), rabbit anti-TRIM21 (D101D, Cell Signalling Technology, #92043; 1:1,000), rabbit anti-NLRP3 (D4D8T, Cell Signalling Technology, #15101; 1:500), goat anti-AdV polyclonal (AB1056, Merck Millipore; 1:500), rabbit anti-GFP (Abcam, ab6556; 1:2,000), rabbit anti-cGAS (D1D3G, CST, 15102, 1:1,000), rabbit anti-STING (D2P2F, CST, 13647, 1:1,000) and rabbit anti-COXIV (LI-COR 926-42214; 1:5,000). HRP-coupled secondary anti-mouse (Dako, P0260), anti-rabbit (Fisher, 31462), anti-goat (Santa Cruz, sc-2056) and anti-β-Actin (Santa Cruz, sc-47778) were detected by enhanced chemiluminescence (Amersham, GE Healthcare) and X-ray films, Alternatively, infrared dye 780 nm or 800 nm coupled secondary antibodies (IRDye 800CW or IRDye 680RD, LI-COR, 1:5,000) were detected using an Odyssey CLx Scanner (LI-COR).

## Genome uptake

Cells were plated at 250,000 cells per well of a 24-well plate and stimulated with AdV complexed with Ab at 50,000 physical particles per cell, in a total volume of 50 μl PBS. 2 h post-infection, cells were washed twice with PBS, harvested and pelleted and gDNA extracted using a DNeasy Blood and Tissue Kit from QIAGEN, according to manufacturer's instructions. Viral DNA was assessed using primers specific to Ad5: Ad5 for TTGCGTCGGTGTTGGAGA, Ad5 rev: AGGC CAAGATCGTGAAGAACC, Ad5 probe: FAM-CTGCACCACATTTCGG CCCCAC-TAMRA by TaqMan qPCR. A dilution series of virus stocks of known concentrations was used as a calibration curve in order to calculate the virus concentration.

## RNA

Human monocyte-derived macrophages were plated at 250,000 cells per well of a 24-well plate and stimulated as indicated. After 3 h, cells were harvested directly in Buffer RLT with β-mercaptoethanol and RNA processed directly using the RNeasy Mini Kit (Qiagen) according to manufacturer's instructions with on-column DNase digestion. RNA concentration was measured using a NanoDrop spectrophotometer and an equal starting concentration used for

| Target | TaqMan primer set |
| --- | --- |
| Human TLR9 | Hs00370913_s1 |
| Human ACTB | Hs01060665_g1 |
| Human IL1B | Hs00174097_m1 |
| Human TNF | Hs00174128_m1 |
| Human IFNB1 | Hs01077958_s1 |
| Human NLRP3 | Hs00366465_m1 |
| Human AIM2 | Hs00915710 |

reverse transcription. Reverse transcription was performed using Superscript II with OligodT 18 priming. Quantitative PCR was performed using relative gene expression compared to β-actin reference gene was determined using the change-in-threshold ($2^{-\Delta\Delta CT}$) method, using TaqMan gene expression assays (Life Technologies) on a StepOnePlus Real Time PCR machine (Applied Biosystems, Waltham, MA). Alternatively, gene expression was determined relative to a standard curve generated from plasmids containing TLR9, AIM2 or actin.

## *In vivo* protocols

Husbandry and housing conditions of experimental animals conform to standards set out under the UK Animals (Scientific Procedures) Act 1986 and the Medical Research Council Animal Welfare and Ethical Review Body. C57BL/6 (JAX:000664) was obtained from Jackson Laboratories. 7–12-week-old males or females (usually 20–30 g) were used in infection experiments, which were conducted in accordance with the 19.b.7 moderate severity limit protocol and Home Office Animals (Scientific Procedures) Act (1986). No weight-matching or sex-matching was performed.

C57/BL6 WT mice were injected i.v. with 100 μl of PBS alone or 100 μl PBS with 2.5 μg hh9C12 WT or hh9C12 H433A antibody. The next day mice were injected i.v. with 100 μl of PBS alone or 100 μl PBS with $2.5 \times 10^{11}$ viral particles AdV-GFP. 4 h after injection, mice were sacrificed and spleens collected. Single cell suspensions from spleens were obtained by mechanically forcing the spleen through nylon 30-μm cell strainers. After red blood cell lysis (Miltenyi), cells were resuspended in ice-cold PBS with 2% FBS and 5 mM EDTA. FC receptors were blocked with FC block (eBioscience, Cat no: 14-0161-81) at 1:100 and then subsequently stained with Cd11b PerCpCy5.5 (BD, Cat no: 550993, 1:100) and Ly6G APC (eBioscince, clone 1A8, Cat no: 17-9668-82, 1:100) and with Dapi (Thermofisher, Cat no: D1306 1:5,000) and measured on a BD LSRFortessa. Data were analysed using FlowJo.

## Acridine orange staining

Human monocyte-derived macrophages (250,000 cells per well of a 24-well plate) were incubated with 5 μg/ml acridine orange (Sigma, Cat no: A8097) for 15 min. Cells were then washed 3× with PBS and then incubated in serum-free RPMI for stimulation with AdV +/− Antibody or LeuLeu O-Me. After 3 h, cells were washed 1× with PBS and then harvested using Trypsin EDTA. For assessing cell death by

eFluor 780 staining (Thermofisher, 65-0865-14), cells were incubated on ice for 15 min in PBS with e780 (1:2,500) prior to fixation.

Cells were subsequently fixed with 4% formaldehyde in PBS (PFA) at room temperature for 15 min. Cells were pelleted and subsequently resuspended in PBS with 2% FBS and 5 mM EDTA for analysis by flow cytometry. Acridine orange red fluorescence was measured on a BD LSR II cytometer, using the 488 nm laser and detecting the red signal using a 585 nm filter.

## Neutralisation assays

Human monocyte-derived macrophages or THP-1s were plated at 50,000 cells per well of a 96-well plate in serum-free RPMI. Virus infectivity was determined in THP-1 cells, and a dilution of virus of 1:200 from a stock of $1.25 \times 10^{12}$ pp/ml was determined as giving a range where 20% cells would be GFP positive, assuming that one GFP-positive cell represents one infectious virus. To assess antibody neutralisation at high doses of virus (that induce cytokine responses), an Ad5-mCherry virus was used in a 1:200 ratio with an Ad5-GFP virus. 2.5 μl of virus was then incubated with 2.5 μl antibody at indicated concentrations and 5 μl PBS. This total 10 μl was added per well. 24 h post-infection HMDM were washed once with PBS then harvested in trypsin and transferred to a 96-well U-Bottomed plate. The cells were then fixed in 4% PFA at room temperature for 15 min. Cells were pelleted and subsequently resuspended in PBS with 2% FBS and 5 mM EDTA for analysis by flow cytometry. GFP or mCherry fluorescence was measured by flow cytometry on the LSRFortessa or the Sony iCyt Eclipse.

## Imaging

### ASC specking

Asc-GFP THP-1s were seeded at 50,000 cells per well of a black, glass-bottomed 96-well plate (Corning, Cat no 354640) and differentiated with PMA. After 48 h, media was changed to serum-free RPMI and zVAD-FMK (InvivoGen, Catalogue no: tlrl-vad) added to a final concentration of 20 μM or H151 added to a final concentration of 5 μM for 30 min before incubation with virus and antibody complexes for 6 h or Nigericin for 1 h. Cells were subsequently fixed in 4% PFA at room temperature for 15 min, then washed once with PBS before adding Hoechst (1:1,000) and Alexa 647 labelled cholera toxin B (CtxB) (1 μg/ml) in PBS. Cells were imaged on the Nikon HCA inverted fluorescence microscope for high content screening using the 20×/0.75NA Air objective. Images were analysed using Nikon NIS Elements Software. Each condition had two wells per plate, with six fields per well in a random field selection, and was analysed as the number ASC specks associated with a cell. Cells were determined as having nuclei (Hoechst stain). The experiment was repeated three times with the final number being the mean of the average number of specking cells across the 12 combined views per treatment in each experiment.

### Co-localisation

Human monocyte-derived macrophages were plated at 250,000 cells per well of a 24-well dish on poly-L Lysine-coated coverslips. Cells were subsequently primed with pI:C for 3 h then stimulated with AdV + antibody complexes or Nigericin for 3 h. Cells were fixed at 37°C for 30 min in fixative (100 mM HEPES pH 7, 50 mM EGTA,

pH 7, 10 mM MgSO$_4$, 2% formaldehyde and 0.2% Triton X-100). Cells were then washed 3–5 times with PBS and 0.1% Triton X-100 (PBT) and then stained with primary antibody (ASC AL177, Adipogen, 1:200) overnight at 4°C. Coverslips were washed 3–5 times in PBT with 3% bovine serum albumin and then incubated with secondary antibodies (Goat anti-Rabbit IgG (H + L) Highly Cross-Adsorbed ThermoFisher Secondary Antibody, Alexa Fluor 488, Cat#A11034 and Goat anti-Human IgG (H + L) Alexa Fluro 647, Cat no. A-21445) for 1 h at room temperature in the dark. Coverslips were washed 3 × 5 times with PBS and then mounted using Pro-Long diamond (Thermofisher, Cat no P36965). Cells were imaged using a Zeiss LSM 710 microscope with a 60× oil objective.

## Data analysis

Statistical analyses were performed using GraphPad Prism 7 software (GraphPad). Error bars depict the SD or SEM as indicated. Data were considered to be statistically significant when $P < 0.05$ by two-tailed Student's $t$-test. In figures, asterisks denote statistical significance as calculated using a Student's $t$-test (*$P < 0.05$; **$P < 0.01$; ***$P < 0.001$; ****$P < 0.0001$).

**Expanded View** for this article is available online.

## Acknowledgements

We gratefully acknowledge Dr Adam Fletcher (MRC–LMB, Cambridge, UK) and Dr Matthew Mangan (Institute of Innate Immunity, University of Bonn, Germany) for helpful discussions. We thank Drs Bernardo Franklin and Eicke Latz (Institute of Innate Immunity, University of Bonn, Germany) for the ASC-GFP THP-1s. Larisa Labzin was supported by an EMBO long-term fellowship (1487-2015) and an NHMRC CJ Martin Fellowship (GNT1124612).

## Author contributions

LIL conceived the project, performed experiments and wrote the manuscript. MB, MV, PR-S and DC performed experiments. SF and JTA provided monoclonal h9C12 antibodies. LCJ supervised the project and wrote the manuscript.

## Conflict of interest

The authors declare that they have no conflict of interest.

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
