## [Review Process File · The EMBO Journal]

Antibody and DNA sensing pathways converge to activate the inflammasome during primary human macrophage infection

Larisa I Labzin, Maria Bottermann, Pablo Rodriguez-Silvestre, Stian Foss, Jan Terje Andersen, Marina Vaysburd, Dean Clift, Leo C James

Review timeline:

Submission date:	13th Dec 2018
Editorial Decision:	11th Feb 2019
Revision received:	29th May 2019
Editorial Decision:	19th Jun 2019
Revision received:	9th Jul 2019
Accepted:	17th Jul 2019

Editor: Karin Dumstrei

Transaction Report:

1st Editorial Decision

11th Feb 2019

Thank you for submitting your manuscript to The EMBO Journal. I am sorry for the delay in getting back to you with a decision, but I have now received the two reports on the manuscript. A third referee who had agreed to review the paper has unfortunately not returned his/her report yet. I have therefore decided to take the decision based upon the two reports at hand.

As you can see below, both referees appreciate the study and find it interesting. However, it is also clear that further work is needed to consider publication here. The referees offer constructive comments for how to extend the analysis. Should you able to add further experiments that would address the raised concerns then I am happy to consider a revised version. I see that point 1 and 2 raised by referee #2 will involve extensive further work, but please note that we don't need the full mechanism, but simply some more insight. Let me know if we need to discuss anything further.

REFEREE REPORTS:

Referee #1:

This manuscript from the James lab reports an interesting and unexpected host-defence pathway, in which antibody coating of adenovirus virions changes the nature of the macrophage response to viral infection. The authors reveal that in addition to neutralizing the virus, viral antibody coating (either with pooled IVIg or with a Adv-specific antibody, 9C12) triggers infected macrophages to activate signaling via the NLRP3 inflammasome pathway, leading to IL-1 production. Intriguingly, the mechanisms leading to NLRP3 activation appear to be distinct depending on the nature of the antibody used to coat the virus - IVIg-coated virus is taken up and then ruptures the lysosome, leading to NLRP3 activation (presumably through similar mechanisms as are described by others for lysosomal rupture agents such as particles, crystals, protein aggregates etc). By contrast, 9C12-

coated virus engages TRIM21, leading to viral capsid destruction and presumably the liberation of viral PAMPs into the cytosol, leading to recognition by cytosolic sensor proteins (e.g. cGAS/STING) and indirect activation of the NLRP3 inflammasome pathway. In sum, this study reveals an exciting new TRIM21 pathway that triggers innate immune sensor function in the macrophage cytosol, by facilitating the liberation and recognition of viral PAMPs. It also reveals an interesting and novel mechanism by which the adaptive immune arm can boost innate immune signaling (particularly interesting because most studies focus on the reverse - innate instruction of adaptive immunity!).

Overall, the study is well-executed, uses appropriate methods, and the manuscript flows logically and is well written. A particular strength of the study is the use of human primary macrophages (and a relevant human myeloid cell line where cell line studies are unavoidable). Inflammasome signaling is enormously relevant to human health, especially given emerging differences in the inflammasome pathways of humans versus model organisms such as mice, and the burgeoning excitement in the potential use of inflammasome inhibitors in the clinic. However inflammasome signaling in humans is much understudied, in part because it requires difficult tools (e.g. CRISPR THP1 lines) to be set up, which the authors have done here and for which they should be congratulated. It is this Reviewer's opinion that this is a strong study worthy of publication in EMBO J, and if the concerns below could be addressed the resulting publication would be excellent and have wide impact in the field. I cannot see any 'fatal flaws' that could preclude publication, and rather, my criticism below is intended to be constructive and to improve the manuscript and its resulting impact.

Recommendation: Moderate revisions.

Major concerns:

1. The molecular basis of the distinct responses of macrophages to AdV coated with IVIg versus 9C12 was well described. It would be helpful if the authors could expand (perhaps in the discussion) on the biological significance of these distinct responses - in the course of a natural infection where anti-AdV antibodies will be part of a pool of circulating antibodies, will the (presumably TRIM21-independent) response dominate over the TRIM21-dependent pathway? Is the IVIg-AdV response fully TRIM21-independent? This was not assessed in Fig. 3E and should be.
2. While the figures themselves were neatly and appropriately set out, they were a little difficult to follow because (1) the figures didn't have titles, and (2) there was sometimes insufficient information in the figure legend to follow exactly what was done to generate the data. For example, in figure 2 where dose escalations of Ab-AdV or Ab per virus were performed, it was difficult to determine from the methods where the equivalent stimulations were in figs A versus B. For the other panels in this figure, the Ab dose is given but not the AdV pp/cell, so it is hard to interpret C, D, E-H in relation to A-B. More information here would be very helpful.
3. Figure 1: Figure 1C western blot is difficult to interpret without the equivalent cell lysate pro-IL-1 blot (and a loading control); please provide these. In figure 1D are the data for AdV versus IVIg AdV (D, upper) and versus AdV + 9C12/IVIg significantly different? They look like they should be, but these statistics were not indicated.
4. Figure 2: The authors state that the 9C12-AdV does not induce HMDM death (Fig 2E), but it does seem to cause some THP1 death (Fig. 2A-B). Related to point 2, are the 9C12-AdV conditions (pp/cell and Ab concentration) in both figures comparable, and if so at which dose in B? I couldn't find this information in the figure legend. Can the differences in cell death responses in HMDM versus THP1 be explained? How exactly were the lysosomal damage experiments in figure 2H performed? The data have a Y axis of % lysosomal damage, which suggests the flow cytometry data was gated somehow to isolate the population of cells with low expression of the red dye. It may be more informative to present this data as MFI in the red channel instead (or as well) - it could be that all cells lose some red (acidic lysosomal) staining with 9C12-AdV, but not enough to separate that cell population into a distinct gate?
5. Figure 3 and associated text: In figure 3E, is TNF similarly dependent on TRIM21, or is this a TRIM21-independent response? Fig. 3F graph should incorporate statistics. In the results section, Fig 3G data is interpreted to mean that "neutralisation of AdV by the same antibody can occur by a multitude of mechanisms", which is incorrect in this Reviewer's opinion. This data indicates that 9C12-coated AdV infects macrophages by two mechanisms that are redundant with one another (pathways requiring the LL residues versus H433 residue, presumably indicating a requirement for FcR versus TRIM21).

6. Figure 4 and associated text: this results section overstates the data. The authors show that 9C12 AdV coating mildly enhances IL-1b expression (unlike the strong upregulation in the polyIC primed control). This is interpreted by the authors to mean that there is no effect of TRIM21 on inflammasome priming. Inflammasome priming is complex and poorly understood, and involves a multiple of mechanisms in addition to the stated pro-IL-1b induction - priming involves upregulation of NLRP3 mRNA/protein, as well as addition/removal of posttranslational modifications such as phosphorylation and ubiquitination. The interpretation of these results should thus be a little more circumspect, e.g. state only that the TRIM21/9C12 pathway is unlikely to enhance IL-1b production via potentiating pro-IL-1b expression. See also non-essential suggestion below.

7. Figure 5 shows convincing data that NLRP3 mediates IL-1b production in response to coated AdV. Cell death responses would also be incredibly informative here. If, as the authors suggest, IVIg-AdV triggers lysosomal rupture, then one would expect that while IL-1b requires NLRP3, cell death would not (in analogy to the other lysosomal rupture agents). If indeed NLRP3 signals in response to 9C12-AdV without concomitant cell death as authors suggest, then there should be no effect of MCC950, K+, or NLRP3/ASC/CASP1 KO on cell death responses to during this infection.

8. Figure 6 data is very interesting, indicating a potential role for cGAS/STING in the IL-1 and TNF responses. Figs 6D, E should be replicated and statistics performed, as the differences are not entirely clear cut. In Fig E, why are nigericin and LPS responses altered in the cGAS/STING KO? This seems odd and is not explained. The authors suggest that cGAS/STING are upstream of NLRP3 activation, but the data in this figure does not actually establish the signaling hierarchy. To do so would be relatively straightforward - ASC specks could be assessed in the cGAS and STING KO (if cGAS and STING are upstream of NLRP3-activated ASC, then these genotypes should be deficient in dispersed speck formation). NFkB activation (e.g. Ikb degradation) and TNF production should however require cGAS and STING whilst NLRP3 should be dispensable. This would establish that cGAS/STING NFkB TNF and also that cGAS/STING NLRP3 activation.

Minor concerns:

9. While generally well written, some passages of the manuscript were confusing or unclear. For example, in the abstract the two sentences starting at lines 15 and 19 seem at odds with one another (one suggests that antibody opsonised AdV triggers lysosomal damage, whilst the latter sentence indicates it does not). In another example, on page 2 line 49 macrophages are stated to have "less acidic lysosomes" but the reader is not told what this comparison is made to. Please also check the placing of commas, as many are mis-placed and decrease clarity.

10. In figure 1, PMA-stimulated THP1s are used. In this case, PMA functions as a priming stimulus, rather than subverting a requirement for priming as is stated in the manuscript.

11. Page 13 discussion line 408 states "antibody-mediated neutralisation occurred in a trim21 and FcgR independent manner" - this seems incorrect. The data suggests that both of these pathways function in a redundant manner to neutralize AdV, such that you only block neutralization when both of these pathways are suppressed.

12. Line 440 discussion: states "it is therefore possible that TRIM21 can activate NLRP3 through an alternative mechanism". This seems unclear/misleading. Suggest instead "...through both cGAS-dependent and -independent mechanisms" for clarity.

Non-essential suggestions for study improvement:

13. Page 47 "H5N1" could be "H5N1 Influenza virus"

14. Page 3 line 68 "via homotypic interactions" suggest interactions between identical proteins. Could be "homotypic domain interactions" or leave out entirely as it is not required to understand the signaling pathway.

15. Page 5 line 146 "only required for" should be "only obligatorily required for" for increased clarity

16. Page 8 line 233: "prevented by H433A mutation" could be "prevented by 9C12 H433A mutation" for clarity, as the subject of this sentence is TRIM21 not 9C12

17. Page 8 line 248 "we infected" could be "we i.v. infected..." for clarity

18. In figure 4, it would be very helpful if NLRP3 mRNA and protein was also measured during the analysis of priming, to determine whether the TRIM21 pathway may upregulate NLRP3 expression and therefore contribute to inflammasome licensing. NLRP3 is known to be modified by ubiquitination, and this can affect its capacity to signal - as an E3 ligase, does TRIM21 modify NLRP3 Ub? This could also be addressed in this figure.

19. The paper repeatedly refers to a "cGAS/STING/NLRP3 inflammasome" which suggests all three proteins come together to form a distinct complex - I don't think this suggestion is supported in the

manuscript or the wider literature. Suggest changing to cGAS/STING/NLRP3 signaling axis or cGAS/STING-induced NLRP3 inflammasome.

Referee #4:

In this work Labzin et al explore the mechanism of inflammasome activation by adenovirus, and the impact of antibody coating. Based on the data presented the authors propose that two pathways are activated, namely lysosome rupture and NLRP3 dependent pyroptosis, and lysosome escape (without cell death) leading to TRIM21 and cGAS dependent NLRP3 activation, which triggers inflammasome activation and NF- κ B activation. Although the proposed model is interesting, the work is underdeveloped, and does at present not fully support the conclusions drawn

1. The proposed link between NLRP3 and NF- κ B is not well characterized, and needs further mechanistic clarification. Since activation of NF- κ B by NLRP3 is not established in the literature, this part needs to be strengthened significantly.
2. Also the link between TRIM21/cGAS and NLRP3 is not thoroughly characterized. According to Gaidt et al, cGAS activation upstream of NLRP3 is mediated by lysosomal cell death and release of K⁺, which acts on adjacent cells. As I read the manuscript, this is not consistent model in the present work, where the authors propose that the TRIM21/cGAS-NLRP3 pathway is not associated with cell death. This needs to be addressed experimentally in more details.
3. Figure 1. Is the observed necrotic cell death pyroptosis? Can it be blocked by caspase 1 inhibition.
4. Figure 1E. It is a relatively low percentage of cells that die at most about 15%. The work would gain if the authors can demonstrate that this is also the cells where extensive lysosomal damage occurs.

General comments.

5. Many data are shown without statistics (e.g 1B, 2A-D). Does this represent singlet observations?
6. The data are presented in multiple formats. For instance, some bar charts are with single data points (e.g. 1H) some are without single data points (e.g. 1E). Is there a reason for this.

1st Revision - authors' response

29th May 2019

Referee #1:

Q1. The molecular basis of the distinct responses of macrophages to AdV coated with IVIg versus h9C12 was well described. It would helpful if the authors could expand (perhaps in the discussion) on the biological significance of these distinct responses - in the course of a natural infection where anti-AdV antibodies will be part of a pool of circulating antibodies, will the (presumably TRIM21-independent) response dominate over the TRIM21-dependent pathway? Is the IVIg-AdV response fully TRIM21-independent? This was not assessed in Fig. 3E and should be.

A1. The question of what antibody responses dominate during different stages of a natural infection is an interesting question and we are grateful to the reviewer for raising it. To address it, we have expanded the discussion to speculate on the biological significance of the different responses (see below). We have also included new data assessing the role of TRIM21 in the AdV-IVIg response, which is now part of new figure 4E. There is a decrease in IL-1 β in TRIM21 KO, suggesting that Adv-IVIg does contain antibodies that can elicit a TRIM21-dependent response. This finding is consistent with our HMDM data, in which we see an IL-1 β response at IVIg doses (0.8 mg/ml) where there is no cell death (Figures 2D, 2G and 2H). Unfortunately, it was not possible to fully separate out the two contributing mechanisms of IL-1 β induction by AdV-IVIg (TRIM21 and lysosomal cell death) because we found the THP-1s to be less sensitive than HMDMs, and that they did release any IL-1 β or TNF at IVIg doses of 0.8mg/ml (Figure 2B, right hand panel).

Taken together, our data suggest that TRIM21 may be particularly important at the early stages of infection where specific antibody titre is low. This would agree with published data on TRIM21 showing that it works at low antibody occupancy (McEwan *et al*, 2012) and is consistent with the importance of the inflammasome during early responses to infection.

Lines 285-287: "IL-1 β release in response to AdV and h9C12 was completely ablated in the Trim21-deficient THP-1s, while AdV-IVIg- (20 mg/ml) induced IL-1 β was reduced."

Lines 560-574: "Quantifying what part of the protective effect of antibodies during AdV infection is due to inflammasome activation and signalling remains to be determined. Of the two mechanisms of antibody-induced inflammasome activation we describe here, we speculate that TRIM21 might be more important at the earlier stages of infection where antibody occupancy on each virion is low and most viruses enter the cell via a natural infectious route. This is suggested by the fact that the TRIM21-mechanism of IL-1 β release is detectable at low antibody doses, whilst IL-1 β release following redirection to the lysosome requires higher antibody occupancy, presumably for efficient Fc γ R cross-linking. This would agree with previous data showing that TRIM21 neutralises AdV at low antibody occupancy (McEwan *et al*, 2012). Antibody paratope may also influence the relative importance of different antibody-induced inflammasome mechanisms, with a dominant anti-hexon repertoire (such as h9C12 used here) more likely to favour a TRIM21-based response."

Q2. While the figures themselves were neatly and appropriately set out, they were a little difficult to follow because (1) the figures didn't have titles, and (2) there was sometimes insufficient information in the figure legend to follow exactly what was done to generate the data. For example, in figure 2 where dose escalations of Ab-AdV or Ab per virus were performed, it was difficult to determine from the methods where the equivalent stimulations were in figs A versus B. For the other panels in this figure, the Ab dose is given but not the AdV pp/cell, so it is hard to interpret C, D, E-H in relation to A-B. More information here would be very helpful.

A2. We apologise for the confusion and have updated both the figure and the figure legends to more accurately present and describe the data. We have also given the figures titles. For clarity we have also separated former Figure 2E-H into a new figure (Figure 3) to accommodate new data.

Q3. Figure 1: Figure 1C western blot is difficult to interpret without the equivalent cell lysate pro-IL-1 blot (and a loading control); please provide these.

A3. We have now included the equivalent cell lysate pro-IL-1 β blots and corresponding b-Actin loading controls in Figure 1C. There is an increase in pro-IL-1 β in all conditions (over time), which suggests that AdV alone is being sensed in THP-1s, but importantly we only see cleaved IL-1 β in AdV + h9C12 or AdV + IVIg conditions, consistent with the ELISA data in Figure 1B.

Lines 156-160: "We confirmed that we were detecting cleaved IL-1 β in the supernatants by western blot (Figure 1C). While AdV alone seemed to enhance expression of pro-IL-1 β protein in PMA differentiated THP-1s, cleaved IL-1 β was only detected in the presence of antibody."

Q4. In figure 1D are the data for AdV versus IVIg AdV (D, upper) and versus AdV + 9C12/IVIg significantly different? They look like they should be, but these statistics were not indicated.

A5. Only those combinations with a statistically significant difference are indicated as such in the data. In the previous dataset, AdV versus IVIg AdV was almost but not quite significant. This was due to the large variability between human donor-derived samples. However, we have now included an additional donor (previously we only had 6 for the IVIg AdV condition) and this increased sample size gives significance when comparing to IL-1 β produced from the same 7 donors challenged with AdV alone ($p = 0.0153$). These 7 donors also showed the same significant increase in IL-1 β release and TNF release upon infection with AdV-h9C12 compared to AdV alone.

Q5. Figure 2: The authors state that the 9C12-AdV does not induce HMDM death (Fig 2E), but it does seem to cause some THP-1 death (Fig. 2A-B). Related to point 2, are the 9C12-AdV conditions (pp/cell and Ab concentration) in both figures comparable, and if so at which dose in B? I couldn't find this information in the figure legend.

A5. We have repeated several of these experiments using a second independent assay to confirm cell death levels and to ensure that the same conditions (eg pp/cell and Ab concentration) are used consistently. For most experiments, we have used 50 000 physical particles (pp) of AdV per cell. We incubated this virus with either 20ug/ml h9C12 or 20 mg/ml IVIg. For the titrations shown in Figure 2A, we have changed the total number of AdV pp we add per cell, but not the amount of

antibody bound per virus. In the rest of Figure 2 and in Figure 3 we have used the same high dose of AdV (50 000 pp/cell) but with a reduced antibody dose, so there are fewer antibodies bound per virion. We have updated the figure legends to more accurately describe these experiments.

We have also optimised the LDH release assay so that it is possible to directly compare cell death between HMDMs and THP-1s. While we observe some cell death in THP-1s with Adv-h9C12 using the LDH assay (Figure 2E), it is not statistically significant – in contrast to using Adv-IVIg. In HMDM, we also only detected significant cell death in the LDH assay with Adv-IVIg (Figure 2G). To independently confirm that the LDH assay was correctly reporting on cell death, we performed additional experiments using two alternative approaches – a PrestoBlue colorimetric assay (Figure 2F, 2H) and a flow cytometry e780 live/dead stain (Figure 3D). In the latter assay, compound e780 is usually cell impermeable, but stains cells with plasma membrane damage. This assay confirmed that Adv +IVIg caused cell death in HMDM, but that Adv alone or Adv + h9C12 did not (Figure 3D). All three assays agree that HMDMs and THP-1s behave similarly, with IVIg causing death but not h9C12.

Crucially, we observe conditions of IL-1 β and TNF release induced by h9C12-Adv where there is no increase in cell death in THP-1 or in HMDM (Figure 2A, B, C). First, in THP-1 (Figure 2A, B) and in HMDM (Figure 2C), Adv alone triggers LDH release but h9C12 does not greatly enhance it (unlike IVIG). Importantly, while the addition of h9C12 does not substantially alter cell death, it nevertheless greatly enhances IL-1 β and TNF release. Moreover, because Adv triggers some LDH release (cell death) without promoting IL-1 β release, it demonstrates that the two events are separable. Second, HMDMs treated with a lower dose of IVIg (50 000 Adv pp/cell with 0.8mg/ml IVIg) release IL-1 β without concomitant cell death (Figure 2D), suggesting that there is a dose where IVIg can act like h9C12. This finding is consistent with published data showing that IL-1 β release can occur independently from cell death in murine bone marrow macrophages and neutrophils (Chen *et al*, 2014)(Monteleone *et al*, 2018)(Evavold *et al*, 2018).

The following new data and discussion has been incorporated into the revised manuscript as follows:

Lines 182-202: "Using the highest dose of virus (50 000 pp/cell), we then measured whether the concentration of antibody per virus impacted cytokine production in THP-1s. Indeed, we found that cytokine release was proportional to the amount of antibody per virus (Figure 2B). We also noted that none of the h9C12 antibody concentrations substantially increased THP-1 cell death compared to virus alone, while the two high doses of IVIg (20 and 4 mg/ml) increased THP-1 cell death. The only dose of Adv-IVIg (0.8 mg/ml) that did not trigger cell death in THP-1s did not trigger any cytokine release in these cells (Figure 2B). We performed the same titrations in HMDM and confirmed that, as in THP-1, h9C12 triggers cytokine release without concomitant cell death (Figure 2C). We also noted that in HMDM the dose of 0.8 mg/ml IVIg triggered cytokine release without the strong increase in cell death seen at the higher IVIg doses, suggesting that in HMDM the separation of cell death and cytokine release with IVIg is also possible (Figure 2D). We confirmed these cell death responses in THP-1s and HMDM by measuring LDH release and by measuring cell viability with a PrestoBlue assay (Figures 2E-H). We saw that only high dose IVIG induced significant cell death in both cell types, and that this occurred independently of pI:C priming (Figure 2G,H). Taken together, this suggests that antibody opsonisation of Adv can trigger IL-1 β or TNF release under conditions that don't trigger concurrent cell death."

Q6. How exactly were the lysosomal damage experiments in figure 2H performed? The data have a Y axis of % lysosomal damage, which suggests the flow cytometry data was gated somehow to isolate the population of cells with low expression of the red dye. It may be more informative to present this data as MFI in the red channel instead (or as well) - it could be that all cells lose some red (acidic lysosomal) staining with 9C12-Adv, but not enough to separate that cell population into a distinct gate?

A6. We have quantified the population of cells with intact lysosomes (red cells) rather than MFI because this gives a larger dynamic window and higher signal to noise. An example of MFI vs gating on the population from a single donor is shown below (bars show mean \pm SD). We have also repeated and expanded this experiment (previously Figure 2H, now Figure 3C) to clarify differences between treatment conditions. Example raw flow cytometry plots have been included in Figure 3C

so that the population shifts and fluorescence can be directly compared. In addition, we have tested an additional two donors, which has given us sufficient power to determine a statistically significant difference in lysosomal damage upon addition of IVIg. Finally, we have made use of the suggestion to quantify MFI in order to assess lysosomal damage specifically in the population of dying cells, as assessed independently by e780 staining (Figure 3D, 3E).

Lines 241-249: "We used the fixable cell viability dye eFluor 780 (e780) to determine whether extensive lysosomal damage is compatible with cell death. Even 4 h post infection, AdV-IVIg at 20 mg/ml caused approximately 20% of cells to die as determined by being positive for the e780 dye (Figure 3D). When we analysed the mean fluorescence intensity (MFI) of the AO red fluorescence in live cells versus dead cells for the AdV-IVIg condition, we found that the e780 positive cells had low AO red fluorescence, suggesting they had undergone extensive lysosomal damage (Figure 3E)."

Q7. Figure 3 and associated text: In figure 3E, is TNF similarly dependent on TRIM21, or is this a TRIM21-independent response?

A7. We have performed the requested experiment and find that TNF release induced by AdV-h9C12 or AdV-IVIg are both highly dependent on TRIM21 in THP-1 cells (New Figure 4F). This is in agreement with the HMDM data presented in Figure 4C, and is consistent with published literature demonstrating TRIM21 activation of NF κ B signalling (McEwan *et al.*, 2013).

Lines 285-295: "IL-1 β release in response to AdV and h9C12 was completely ablated in the Trim21-deficient THP-1s, while AdV-IVIg- (20 mg/ml) induced IL-1b was reduced. Trim21 deficiency did not affect cytokine release in response to transfected double stranded DNA (HT-DNA) or to the NLRP3 agonist Nigericin (Figure 4E). The TNF response to both AdV-h9C12 and AdV-IVIg was also completely ablated in the TRIM21-deficient THP-1s (Figure 4F), though a slight defect in HT-DNA induced TNF responses was also observed, which could indicate further roles for TRIM21 in regulating pro-inflammatory responses. We could not investigate whether TRIM21-deficiency would inhibit low dose IVIg induced cytokine release (as in the HMDM), as this dose did not trigger IL-1 β or TNF release in THP-1s (Figure 2B)."

Q8. Fig. 3F graph should incorporate statistics.

A8. We have added statistics to the graph in Figure 4G (formerly Figure 3F).

Q9. In the results section, Fig 3G data is interpreted to mean that "neutralisation of AdV by the same antibody can occur by a multitude of mechanisms", which is incorrect in this Reviewer's opinion. This data indicates that 9C12-coated AdV infects macrophages by two mechanisms that are

redundant with one another (pathways requiring the LL residues versus H433 residue, presumably indicating a requirement for FcR versus TRIM21).

A9. We have clarified the text in this section as follows:

Lines 313-325: "AdV infection decreased with increasing h9C12 antibody concentration, indicating dose-dependent neutralization (Figure 4H). Using H433A h9C12, which ablates TRIM21 binding, did not prevent neutralization. This is in contrast to non-immune cells such as 293Ts where h9C12 neutralisation of AdV is completely TRIM21 dependent (Bottermann *et al*, 2016). Using a LALA h9C12 with reduced Fc γ R binding also failed to prevent neutralization. However, when H433A was combined with the LALA mutations to inhibit binding to both Trim21 and the Fc γ Rs, neutralisation was prevented at intermediate antibody concentrations (Figure 4H). This suggests that there are multiple redundant routes of viral entry, in which virus:antibody complexes can engage with either Fc γ Rs or TRIM21. This is consistent with a requirement for multiple inflammasome activation mechanisms to detect infection."

Q10. Figure 4 and associated text: this results section overstates the data. The authors show that 9C12 AdV coating mildly enhances IL-1 β expression (unlike the strong upregulation in the polyIC primed control). This is interpreted by the authors to mean that there is no effect of TRIM21 on inflammasome priming. Inflammasome priming is complex and poorly understood, and involves a multiple of mechanisms in addition to the stated pro-IL-1 β induction - priming involves upregulation of NLRP3 mRNA/protein, as well as addition/removal of posttranslational modifications such as phosphorylation and ubiquitination. The interpretation of these results should thus be a little more circumspect, e.g. state only that the TRIM21/9C12 pathway is unlikely to enhance IL-1 β production via potentiating pro-IL-1 β expression. See also non-essential suggestion below.

A10. We agree that the previous summary of these results was an over-simplification and have amended the text as suggested (see below). We have also included new data showing that neither NLRP3 mRNA (Figure 5C) nor NLRP3 protein (Figure 5D) was upregulated by AdV-h9C12 or AdV-IVIg in HMDM. This suggests that TRIM21 is not enhancing inflammasome priming by increasing expression of NLRP3.

Lines 348-353: "The NLRP3 inflammasome is also regulated in part by NF κ B- dependent transcription of NLRP3. However we found that neither AdV-h9C12 nor AdV-IVIg induced NLRP3 mRNA (Figure 5C) or protein expression (Figure 5D). This indicates that the h9C12/TRIM21 pathway is unlikely to enhance IL-1 β production via potentiating either pro-IL-1 β or NLRP3 expression."

Q11. Figure 5 shows convincing data that NLRP3 mediates IL-1 β production in response to coated Adv. Cell death responses would also be incredibly informative here. If, as the authors suggest, IVIg-Adv triggers lysosomal rupture, then one would expect that while IL-1 β requires NLRP3, cell death would not (in analogy to the other lysosomal rupture agents). If indeed NLRP3 signals in response to 9C12-Adv without concomitant cell death as authors suggest, then there should be no effect of MCC950, K+, or NLRP3/ASC/CASP1 KO on cell death responses to during this infection.

A11. We are grateful to the reviewer for this suggestion as it provided an excellent test of our proposed mechanism and that the lysosomal rupture and cell death induced by Adv + IVIg is upstream of NLRP3 rather than caused by NLRP3 activation. We therefore performed additional experiments measuring cell death responses in HMDM in the presence of either the NLRP3 inhibitor MCC950 (Figure 6C) or the Caspase 1 inhibitor VX-765 (Figure 6E). We observed that whilst neither MCC950 nor VX-765 affected Adv + IVIg induced cell death, both inhibitors impaired IL-1 β release (Figures 6A and 6E). We also observed no effect of either MCC950 or VX-765 on cell death during Adv-h9C12 treatment. This data is consistent with a mechanism of NLRP3 signalling without a concomitant increase in cell death.

Lines 380-391: "In HMDM, IL-1 β release induced by Adv and h9C12 or IVIg was ablated by the specific NLRP3 inhibitor MCC950(Coll *et al*, 2015), suggesting that NLRP3 is required for both TRIM21 dependent, and independent, antibody induced IL-1 β release (Figure 6A). As expected, TNF release was not inhibited by MCC950 (Figure 6B). Adv-IVIg induced cell death was not

inhibited by MCC950, suggesting it is not pyroptosis (Figure 6C). Consistent with this result, NLRP3-, ASC- and caspase-1- deficient THP-1s no longer released IL-1 β in response to Adv and antibody (Figure 6D). Furthermore, inhibition of caspase-1 by the inhibitor VX-765 inhibited IL-1 β release in HMDM but not Adv-IVIg induced cell death (Figure 6E), consistent with effects seen with MCC950."

Q12. Figure 6 data is very interesting, indicating a potential role for cGAS/STING in the IL-1 and TNF responses. Figs 6D, E should be replicated and statistics performed, as the differences are not entirely clear cut. In Fig E, why are nigericin and LPS responses altered in the cGAS/STING KO? This seems odd and is not explained.

A12. We have repeated our experiments in cGAS or STING KO THP-1 cells as requested and combined data from three independent experiments, expressed as either the absolute protein values or the % of cytokine normalised to the Ctrl cell line response (Figure 7D). We reproducibly see a significant reduction in IL-1 β in response to Adv + h9C12 in the cGAS and STING KO cells, and a reduction in TNF in the cGAS KO. The repeat experiments have also helped to reveal that there is not a significant difference in the TNF or IL-1 β responses to LPS or Nigericin.

To provide further support for this section of our manuscript, we have also added a second independent set of experiments to test the involvement of cGAS/STING in the antibody-induced IL-1 β response. We made use of the newly described STING inhibitor H151 (Haag *et al*, 2018) in WT THP-1 cells challenged with Adv + h9C12. In agreement with published literature (Haag *et al*, 2018; Gaidt *et al*, 2017), H151 inhibited HT-DNA and cGAMP induced IL-1 β and TNF, while Nigericin-induced IL-1 β responses were unaffected (Figure 7E). Importantly however, Adv + h9C12 induction of IL-1 β was reduced by H151 (Figure 7E). This new data is consistent with the results obtained in KO cells and supports the involvement of STING in the Adv+h9C12 induction of IL-1 β . The H151 compound did not reduce TNF signalling via Adv+h9C12 (Figure 7E), unlike the complete STING KO, presumably because 100% inhibition is not achieved and this pathway has a lower activation threshold compared to IL-1 β . This result may also be due to the ability of TRIM21 to induce TNF transcription by directly activating NF κ B signalling, as previously described (McEwan *et al*, 2013). The new data is presented in Figure 7E, including both a representative experiment with absolute protein levels and an average of multiple experiments where statistical significance for IL-1 β is demonstrated.

Lines 440–453: "We observed that IL-1 β and TNF release in response to Adv and h9C12 was reduced in cGAS and STING- deficient THP-1s (Figure 7D). As expected, HT-DNA responses were impaired, and Nigericin and LPS responses in these knockout THP-1s were unaffected. We then utilised the newly described STING inhibitor H151 (Haag *et al*, 2018) to further investigate this pathway. We found that H151 very efficiently inhibited HT-DNA and cGAMP induced IL-1 β and TNF responses in WT THP-1s without impacting Nigericin induced IL-1 β , showing that it specifically inhibited the STING response (Figure 7E). Adv-h9C12 induced IL-1 β was inhibited by H151, suggesting that cGAS and STING are indeed involved in this pathway. However for Adv-IVIg IL-1 β release was less inhibited, consistent with a role for early lysosomal damage directly triggering NLRP3 activation. Unexpectedly, Adv-h9C12 and Adv-IVIg induced TNF responses were not inhibited by H151 (Figure 7E)."

Q13. The authors suggest that cGAS/STING are upstream of NLRP3 activation, but the data in this figure does not actually establish the signaling hierarchy. To do so would be relatively straightforward - ASC specks could be assessed in the cGAS and STING KO (if cGAS and STING are upstream of NLRP3-activated ASC, then these genotypes should be deficient in dispersed speck formation).

A13. We are grateful for this suggestion and have performed additional experiments to show that cGAS/STING are upstream of NLRP3 in the signalling hierarchy, with ASC-Specking dependent on cGAS/STING signalling. We utilised the STING inhibitor H151 to treat our THP-1- ASC-Cer cells and found that it prevented ASC specking in response to Adv + h9C12 (Figure 7F). Consistent with the IL-1 β responses shown in Figure 7E, H151 did not substantially alter ASC Specking induced by Adv + IVIg. This data is again consistent with the distinct mechanisms of inflammasome activation induced by h9C12 and IVIg. A representative image of an ASC specking experiment and quantification of 3 repeats is shown in Figure 7F.

Lines 453-455: “Finally, H151 also inhibited AdV-h9C12 induced ASC-Specking in THP-1s expressing ASC-GFP (Figure 7F), indicating that STING is upstream of NLRP3 activation.”

Q14. NFκB activation (e.g. IκB degradation) and TNF production should however require cGAS and STING whilst NLRP3 should be dispensable. This would establish that cGAS/STING NFκB TNF and also that cGAS/STING NLRP3 activation.

A14. We agree that this should be the case and indeed we see that NLRP3 inhibition with MCC950 or KCl doesn't affect TNF responses to AdV + h9C12 in HMDM (Figure 6B, 6F). Conversely, Nigericin-induced IL-1β release was not impacted in cGAS and STING deficient THP-1s, or when using the STING inhibitor H151 in WT THP-1s (Figures 7D, 7E). This shows that the cGAS/STING-NFκB signalling axis and the cGAS/STING-induced NLRP3 inflammasome are separable pathways. However, while we observe that AdV+ h9C12 induced TNF release is not inhibited by H151 (Figure 7E), we expect that this is because NFκB can also be activated directly by TRIM21 (Watkinson *et al*, 2015)(McEwan *et al*, 2013).

Minor concerns:

We thank the reviewer for highlighting these points and we have addressed them in the text as follows:

Q15. While generally well written, some passages of the manuscript were confusing or unclear. For example, in the abstract the two sentences starting at lines 15 and 19 seem at odds with one another (one suggests that antibody opsonised AdV triggers lysosomal damage, whilst the latter sentence indicates it does not). In another example, on page 2 line 49 macrophages are stated to have "less acidic lysosomes" but the reader is not told what this comparison is made to. Please also check the placing of commas, as many are mis-placed and decrease clarity.

A15. We have clarified the revised manuscript as suggested. Specifically:

Lines 21-28: “Our data demonstrate that antibody opsonization of virions can activate macrophages in multiple ways. In the first, antibody binding of adenovirus causes lysosomal damage, activating NLRP3 to drive inflammasome formation and IL-1β release. Importantly, this mechanism enhances virion capture but not infection and is accompanied by cell death, denying the opportunity for viral replication. Unexpectedly, we also find that antibody-coated viruses, which successfully escape into the cytosol trigger a second system of inflammasome activation.”

Lines 60-63: “Macrophages are reported to have less acidic endosomes than epithelial cells(Marvin *et al*, 2017) (and references therein), potentially hampering infection as viruses use low pH environments as cues to trigger fusion or membrane permeabilization. ”

Q16. In figure 1, PMA-stimulated THP-1s are used. In this case, PMA functions as a priming stimulus, rather than subverting a requirement for priming as is stated in the manuscript.

A16: We have adjusted the text to reflect this important distinction.

Lines 164-169: “Importantly, IL-1β production was only seen in HMDM primed with the TLR3 agonist poly I:C (pI:C), consistent with two-step inflammasome activation and in contrast to the response in THP-1s, where PMA differentiation is also sufficient as a priming stimulus to drive expression of pro-IL-1β(Dostert *et al*, 2008) and where AdV alone enhances pro-IL-1β expression (Figure 1C).”

Q17. Page 13 discussion line 408 states "antibody-mediated neutralisation occurred in a trim21 and FcγR independent manner" - this seems incorrect. The data suggests that both of these pathways function in a redundant manner to neutralize AdV, such that you only block neutralization when both of these pathways are suppressed.

A17. This has been changed as follows:

Lines 485-490: "...in primary macrophages we saw that inhibiting either TRIM21 or FcγR interactions did not impact antibody mediated neutralisation. However, when both pathways were inhibited antibody mediated neutralisation was compromised. This suggests that the TRIM21 and FcγR pathways function in a redundant manner to neutralise AdV, such that you only block neutralisation when both of these pathways are suppressed."

Q18. Line 440 discussion: states "it is therefore possible that TRIM21 can activate NLRP3 through an alternative mechanism". This seems unclear/misleading. Suggest instead "...through both cGAS-dependent and -independent mechanisms" for clarity.

A18. We have amended the text accordingly.

Lines 521-522: "It is therefore possible that TRIM21 can activate NLRP3 through both cGAS/STING dependent and independent mechanisms."

Q19. Page 47 "H5N1" could be "H5N1 Influenza virus".

A19: We have amended the text accordingly.

Lines 56-58: "Highly pathogenic H5N1 influenza virus replicates in both dendritic cells (DCs) and macrophages(Westenius *et al*, 2018)..."

Q20. Page 3 line 68 "via homotypic interactions" suggest interactions between identical proteins. Could be "homotypic domain interactions" or leave out entirely as it is not required to understand the signaling pathway.

A20: Thank you for this suggestion for clarity. We have decided to remove this entirely for clarity.

Line 80-81: "...which subsequently recruit the adaptor protein ASC and the protease caspase-1."

Q21. Page 5 line 146 "only required for" should be "only obligatorily required for" for increased clarity.

A21. Line 169-171: "Finally, while priming in HMDM was obligatory for antibody-induced IL-1β secretion, TNF secretion was also induced by antibody-virion complexes even in un-primed HMDM (Figure 1D)."

Q22. Page 8 line 233: "prevented by H433A mutation" could be "prevented by 9C12 H433A mutation" for clarity, as the subject of this sentence is TRIM21 not 9C12.

A22: We have adjusted all the text so that '9C12' precedes any description of the mutation in the text. Moreover, as this is recombinant mouse-human chimeric 9C12, we have now adjusted the text to refer to this antibody as the more accurate h9C12 throughout.

Line 275-278: "In response to opsonized AdV, TRIM21 mediates pro-inflammatory cytokine expression and this is prevented by h9C12-H433A mutation(McEwan *et al*, 2013)."

Q23. Page 8 line 248 "we infected" could be "we i.v. infected" ..." for clarity.

A23. Line 300-301: "...in vivo in a TRIM21-dependent manner, we i.v. infected WT mice with virus..."

Q24. In figure 4, it would be very helpful if NLRP3 mRNA and protein was also measured during the analysis of priming, to determine whether the TRIM21 pathway may upregulate NLRP3 expression and therefore contribute to inflammasome licensing. NLRP3 is known to be modified by ubiquitination, and this can affect its capacity to signal - as an E3 ligase, does TRIM21 modify NLRP3 Ub? This could also be addressed in this figure.

A24. We have performed the suggested experiment, which shows that there is no marked upregulation of NLRP3 mRNA 3 h post infection with AdV + Ab (Figure 5C) or upregulation in

NLRP3 protein after 6h (Figure 5D). This suggests that TRIM21 does not play a significant role in promoting licensing by increasing NLRP3 expression and is consistent with our other data that a stronger NF κ B signal is required. We also did not find any direct colocalisation between antibody-virus complexes and ASC, which suggests that there is also no direct colocalisation (or interaction) between TRIM21 and NLRP3 (Figure 6G). Given the antibody-dependence of TRIM21 activation and substrate recruitment we think it unlikely that TRIM21 is an NLRP3-specific ligase.

Lines 348-353: "The NLRP3 inflammasome is also regulated in part by NF κ B- dependent transcription of NLRP3. However, we found that neither AdV-h9C12 nor AdV-IVIg induced NLRP3 mRNA (Figure 5C) or protein expression (Figure 5D). This indicates that the h9C12/TRIM21 pathway is unlikely to enhance IL-1 β production via potentiating either pro-IL-1 β or NLRP3 expression."

Q25. The paper repeatedly refers to a "cGAS/STING/NLRP3 inflammasome" which suggests all three proteins come together to form a distinct complex - I don't think this suggestion is supported in the manuscript or the wider literature. Suggest changing to cGAS/STING/NLRP3 signaling axis or cGAS/STING-induced NLRP3 inflammasome.

A25. We have made this change as suggested and refer to it throughout the text as the cGAS/STING- induced NLRP3 inflammasome.

Referee #4:

In this work Labzin et al explore the mechanism of inflammasome activation by adenovirus, and the impact of antibody coating. Based on the data presented the authors propose that two pathways are activated, namely lysosome rupture and NLRP3 dependent pyroptosis, and lysosome escape (without cell death) leading to TRIM21 and cGAS dependent NLRP3 activation, which triggers inflammasome activation and NF- κ B activation. Although the proposed model is interesting, the work is underdeveloped, and does at present not fully support the conclusions drawn.

Q1. The proposed link between NLRP3 and NF- κ B is not well characterized, and needs further mechanistic clarification. Since activation of NF- κ B by NLRP3 is not established in the literature, this part needs to be strengthened significantly.

A1. We have clarified the signalling hierarchies of the two distinct antibody-dependent mechanisms of macrophage activation we describe in our manuscript. Some of this new data is discussed in answers to Reviewer 1 questions 12-14. A key approach of our study is the quantification of two complimentary but distinct immune pathways that are activated in macrophages during viral infection: inflammasome activation (as measured by IL-1 β release) and NF κ B activation (as measured by TNF release). NF κ B signalling is known to be required for IL-1 β release, as it is required for expression of pro-IL-1 β . However, we do not suggest that NLRP3 activates NF κ B directly; indeed we show that NLRP3 inhibition by the inhibitor MCC950 does not prevent induction of TNF (Figure 6B) by AdV + h9C12, indicating that NLRP3 is not required for NF κ B signalling. Similarly, inhibiting K⁺ efflux-mediated NLRP3 activation with high extracellular KCl blocks IL-1 β release in response to AdV-h9C12 and dsDNA but does not impact TNF release (Figure 6F).

Importantly, the involvement of TRIM21, cGAS and STING in the activation of the NLRP3 inflammasome is independent of their ability to induce NF κ B signalling. They are all required for NLRP3 activation in cells challenged with AdV + h9C12 that have undergone prior NF κ B dependent licensing. For example, TRIM21 is required for AdV-h9C12 induced IL-1 β release even in HMDM with prior NF κ B stimulation (Figure 4B). In the TRIM21-dependent mechanism of IL-1 β release we have uncovered, the adenovirus capsid protecting the viral DNA is degraded and the cGAS/STING- dependent NLRP3 inflammasome is activated (Gaidt *et al*, 2017).

It is possible that IL-1 β , released in an NLRP3 dependent manner, could activate NF κ B in neighbouring cells by signalling through its cognate receptor, however, we would like to emphasise that we do not suggest anywhere in this study that NLRP3 can directly activate NF κ B.

Q2. Also the link between TRIM21/cGAS and NLRP3 is not thoroughly characterized. According to Gaidt et al, cGAS activation upstream of NLRP3 is mediated by lysosomal cell death and release of K⁺, which acts on adjacent cells. As I read the manuscript, this is not consistent model in the present work, where the authors propose that the TRIM21/cGAS-NLRP3 pathway is not associated with cell death. This needs to be addressed experimentally in more details.

A2. NLRP3 activation is linked to the loss of intracellular potassium (efflux), rather than K⁺ uptake by an adjacent cell (Gaidt *et al*, 2017). If uptake of K⁺ were to stimulate NLRP3 then the addition of KCL to cell media would activate the inflammasome. In fact, the addition of KCL inhibits IL-1 β release and is commonly used to confirm K⁺ efflux as the mechanism of NLRP3 activation. For instance, we used high extracellular KCl to show inhibition of IL-1 β release during AdV+ h9C12 and AdV + IVIg challenge, consistent with their activation of the inflammasome via NLRP3 (Figure 6F).

The mechanism for cGAS/STING activation of NLRP3 described by Gaidt et al is that STING re-localisation causes lysosomal destabilisation and subsequent damage to the plasma membrane, with the associated loss of intracellular K⁺ being sensed by NLRP3 to trigger inflammasome activation and IL-1 β release. This is analogous to NLRP3 activation downstream of detection of intracellular LPS by Casp11/4/5. Caspase 11/4/5 are activated upon binding to cytosolic LPS, whereupon they can cleave GSDMD, but not IL-1 β . The resulting GSDMD pores drive pyroptosis and loss of intracellular K⁺, leading to NLRP3 activation and IL-1 β release from the dying cell (Baker *et al*, 2015).

However, while cell death is observed along with IL-1 β release in the Gaidt et al study, whether IL-1 β release can occur *without* cell death in response to detection of cytosolic DNA by cGAS/STING has not been addressed. Recent papers have suggested that IL-1 β can be released from living cells, including macrophages (Monteleone *et al*, 2018; Evavold *et al*, 2018) and neutrophils (Chen *et al*, 2014)(Karmakar *et al*, 2015). These differences can be ascribed to a number of factors including the strength of the response, type and concentration of stimuli and cell type. Our model for AdV-h9C12 is consistent with cGAS/STING activating NLRP3 by inducing K⁺ efflux after lysosomal damage but suggests that cell death is not required.

That cell death is not a pre-requisite for IL-1 β release is also consistent with growing evidence that pyroptosis can be halted and reversed by the ESCRT dependent membrane repair machinery, which can resolve GSDMD pores (Rühl *et al*, 2018). More specifically, it has also been shown that the ESCRT pathway can repair lysosomal membrane damage and promote cell survival (Radulovic *et al*, 2018). Therefore, while AdV + h9C12 does not induce measurable lysosomal cell death it does not preclude lysosomal damage, resulting in K⁺ efflux and IL-1 β release, but with subsequent membrane repair.

To directly test whether it is possible to activate IL-1 β via the DNA sensing pathway in the absence of cell death, we have performed additional experiments closely monitoring both IL-1 β and LDH release. This new data shows that there are doses of either the cGAS ligand, DNA, or the STING binding second messenger, cGAMP, that can trigger IL-1 β induction without a significant increase in cell death compared to lipofectamine alone, unlike the direct NLRP3 agonist Nigericin. (Data is in THP-1s, 16h post transfection, and shows the Average \pm SD of duplicate readings and is one experiment representative of two.)

Taken together, our data is consistent with AdV+h9C12 activation of the cGAS/STING/NLRP3 pathway of IL-1 β release without dramatic cell death.

Q3. Figure 1. Is the observed necrotic cell death pyroptosis? Can it be blocked by caspase 1 inhibition?

A3. We thank the reviewer for this suggestion. We have performed the suggested experiment and tested the effect of caspase-1 inhibitor VX-765 in HMDMs. We observed that VX-765 did not inhibit AdV + IVIg induced cell death as measured by LDH release but that it did inhibit IL-1 β release (Figure 6E), in keeping with its role in inhibiting IL-1 β processing by caspase 1. This suggests that the cell death caused by AdV + IVIg is unlikely to be caspase-1 mediated pyroptosis. This is also in agreement with other new data we have added, using the NLRP3 inhibitor MCC950, where we also did not see any effect on AdV + IVIg induced cell death (Figure 6C). As NLRP3 is upstream of caspase 1, this data supports the interpretation that the observed cell death is caspase 1 independent. We have corrected the manuscript to incorporate this new data and amend the description to cell death.

Lines 385-391: "AdV-IVIg induced cell death was not inhibited by MCC950, suggesting it is not pyroptosis (Figure 6C). Consistent with this result, NLRP3-, ASC- and caspase-1- deficient THP-1s no longer released IL-1 β in response to AdV and antibody (Figure 6D). Furthermore, inhibition of caspase-1 by the inhibitor VX-765 inhibited IL-1 β release in HMDM but not AdV-IVIg induced cell death (Figure 6E), consistent with effects seen with MCC950."

Lines 512-515: "In contrast, we found in HMDMs that the cell death accompanying IL-1 β release in response to AdV-IVIg was unlikely to be pyroptosis, as it was not inhibited by MCC950 or VX-765 (Figure 6C, E), and that for h9C12, IL-1 β release occurred without significant cell death."

Q4. Figure 1E. It is a relatively low percentage of cells that die at most about 15%. The work would gain if the authors can demonstrate that this is also the cells where extensive lysosomal damage occurs.

A4. We have performed additional experiments to address this question and related points raised by the previous reviewer. Upon optimisation of the LDH assay for HMDMs we find that % LDH release is reproducibly closer to 40% during AdV-IVIg stimulation (Figure 2G). To confirm this result we have also performed independent PrestoBlue assays for cell death (Figure 2H), and used e780 live/dead staining to quantify dying cells by flow cytometry (Figure 3D). As in the other assays, AdV + IVIg 20 mg/ml was the only condition in which extensive cell death was observed. To address whether these dying cells are also the cells where extensive lysosomal damage occurs, we took advantage of our new e780 approach and combined it with an Acridine Orange assay to measure lysosomal damage and cell death within the same cells. We first defined our cells as 'living' or 'dying' based on their permeability to the e780 stain (Figure 3D). We then analysed the mean fluorescence intensity of the Acridine Orange red fluorescence in either the 'live' or the 'dead' populations for the AdV + IVIg 20 condition. We observed a loss of AO red fluorescence in dying cells, suggesting that the dying cells have indeed undergone extensive lysosomal damage (Figure 3E). This is presented as part of Figure 3D-E.

Lines 241-249: “We used the fixable cell viability dye eFluor 780 (e780) to determine whether extensive lysosomal damage is compatible with cell death. Even 4 h post infection, AdV-IVIg at 20 mg/ml caused approximately 20% of cells to die as determined by being positive for the e780 dye (Figure 3D). When we analysed the mean fluorescence intensity (MFI) of the AO red fluorescence in live cells versus dead cells for the AdV-IVIg condition, we found that the e780 positive cells had low AO red fluorescence, suggesting they had undergone extensive lysosomal damage (Figure 3E).”

General comments:

Q5. Many data are shown without statistics (e.g. 1B, 2A-D). Does this represent singlet observations?

A5. Where data is shown without statistics, this is either because the observations were not significant, or because the data is a single experiment representative of multiple repeats. We have now clarified this in the updated figure legends. Where possible we have combined repeat experiments to perform statistics. For single experiments, cells were typically treated in duplicate, so the average reading + SD is shown.

Q6. The data are presented in multiple formats. For instance, some bar charts are with single data points (e.g. 1H) some are without single data points (e.g. 1E). Is there a reason for this?

A6. We have tried where possible to show HMDM data as bar charts with each data point representing an individual donor to give a clear indication of donor to donor variability. Where we have used cell lines, dose responses or timecourses we have used line graphs for clarity, unless directly comparing to HMDMs.

References:

- Baker PJ, Boucher D, Bierschenk D, Tebartz C, Whitney PG, D’Silva DB, Tanzer MC, Monteleone M, Robertson AAB, Cooper MA, Alvarez-Diaz S, Herold MJ, Bedoui S, Schroder K & Masters SL (2015) NLRP3 inflammasome activation downstream of cytoplasmic LPS recognition by both caspase-4 and caspase-5. *Eur. J. Immunol.* **45**: 2918–2926
- Bottermann M, Lode HE, Watkinson RE, Foss S, Sandlie I, Andersen JT & James LC (2016) Antibody-antigen kinetics constrain intracellular humoral immunity. *Sci. Rep.* **6**: 37457
- Chen KW, Groß CJ, Sotomayor FV, Stacey KJ, Tschopp J, Sweet MJ & Schroder K (2014) The Neutrophil NLRP3 Inflammasome Selectively Promotes IL-1 β Maturation without Pyroptosis during Acute Salmonella Challenge. *Cell Rep.* **8**: 570–582
- Coll RC, Robertson AAB, Chae JJ, Higgins SC, Muñoz-Planillo R, Inserra MC, Vetter I, Dungan LS, Monks BG, Stutz A, Croker DE, Butler MS, Haneklaus M, Sutton CE, Núñez G, Latz E, Kastner DL, Mills KHG, Masters SL, Schroder K, et al (2015) A small-molecule inhibitor of the NLRP3 inflammasome for the treatment of inflammatory diseases. *Nat. Med.* **21**: 248–255
- Dostert C, Petrilli V, Van Bruggen R, Steele C, Mossman BT & Tschopp J (2008) Innate Immune Activation Through Nalp3 Inflammasome Sensing of Asbestos and Silica. *Science.* **320**: 674–677
- Evavold CL, Ruan J, Tan Y, Xia S, Wu H & Kagan JC (2018) The Pore-Forming Protein Gasdermin D Regulates Interleukin-1 Secretion from Living Macrophages. *Immunity* **48**: 35-44.e6
- Gaidt MM, Ebert TS, Chauhan D, Ramshorn K, Pinci F, Zuber S, O’Duill F, Schmid-Burgk JL, Hoss F, Buhmann R, Wittmann G, Latz E, Subklewe M & Hornung V (2017) The DNA Inflammasome in Human Myeloid Cells Is Initiated by a STING-Cell Death Program Upstream of NLRP3. *Cell* **171**: 1110-1124.e18
- Haag SM, Gulen MF, Reymond L, Gibelin A, Abrami L, Decout A, Heymann M, van der Goot FG, Turcatti G, Behrendt R & Ablasser A (2018) Targeting STING with covalent small-molecule inhibitors. *Nature* **559**: 269–273
- Karmakar M, Katsnelson M, Malak HA, Greene NG, Howell SJ, Hise AG, Camilli A, Kadioglu A, Dubyak GR & Pearlman E (2015) Neutrophil IL-1 β Processing Induced by Pneumolysin Is Mediated by the NLRP3/ASC Inflammasome and Caspase-1 Activation and Is Dependent on K⁺ Efflux. *J. Immunol.* **194**: 1763–1775
- Marvin SA, Russier M, Huerta CT, Russell CJ & Schultz-Cherry S (2017) Influenza Virus Overcomes Cellular Blocks To Productively Replicate, Impacting Macrophage Function. *J. Virol.* **91**: e01417-16

- McEwan WA, Hauler F, Williams CR, Bidgood SR, Mallery DL, Crowther RA & James LC (2012) Regulation of Virus Neutralization and the Persistent Fraction by TRIM21. *J. Virol.* **86**: 8482–8491
- McEwan WA, Tam JCH, Watkinson RE, Bidgood SR, Mallery DL & James LC (2013) Intracellular antibody-bound pathogens stimulate immune signaling via the Fc receptor TRIM21. *Nat. Immunol.* **14**: 327–336
- Monteleone M, Stanley AC, Chen KW, Brown DL, Bezbradica JS, von Pein JB, Holley CL, Boucher D, Shakespear MR, Kapetanovic R, Rolfes V, Sweet MJ, Stow JL & Schroder K (2018) Interleukin-1 β Maturation Triggers Its Relocation to the Plasma Membrane for Gasdermin-D-Dependent and -Independent Secretion. *Cell Rep.* **24**: 1425–1433
- Radulovic M, Schink KO, Wenzel EM, Nähse V, Bongiovanni A, Lafont F & Stenmark H (2018) ESCRT-mediated lysosome repair precedes lysophagy and promotes cell survival. *EMBO J.* **37**: e99753
- Rühl S, Demarco B, Heilig R, Santos JC, Broz P & Shkarina K (2018) ESCRT-dependent membrane repair negatively regulates pyroptosis downstream of GSDMD activation. *Science.* **362**: 956–960
- Watkinson RE, McEwan WA, Tam JCH, Vaysburd M & James LC (2015) TRIM21 Promotes cGAS and RIG-I Sensing of Viral Genomes during Infection by Antibody-Opsonized Virus. *PLOS Pathog.* **11**: e1005253
- Westenius V, Mäkelä SM, Julkunen I & Österlund P (2018) Highly Pathogenic H5N1 Influenza A Virus Spreads Efficiently in Human Primary Monocyte-Derived Macrophages and Dendritic Cells. *Front. Immunol.* **9**: 1664

2nd Editorial Decision

19th Jun 2019

Thank you for submitting your revised version to The EMBO Journal. Your revision has now been seen by the two referees and both appreciate the introduced changes. I am therefore happy to let you know that we are pleased to accept the publication of your study here. Before I can see you the formal acceptance letter there are just a few minor things to sort out

REFeree REPORTS:

Referee #1:

The authors have nicely addressed all of my previously concerns, and I congratulate them on a very interesting paper.

One minor concern (missed previously, apologies): pro-IL-1 β appears to be running at around 40kDa in the blots, but should be lower than the 35 kDa marker (it is 31 kDa). Is it possible that the marker has been incorrectly labelled? Please check this.

Referee #4:

The authors have done an impressive job addressing the points raised, and I am now convinced that the conclusions drawn are supported by the data.

2nd Revision - authors' response

9th Jul 2019

The authors performed all requested editorial changes.

3rd Editorial Decision

17th Jul 2019

Thank you for submitting your revised manuscript to The EMBO Journal. I have now had a chance to take a look at everything and all looks good. I am therefore very pleased to accept the manuscript for publication here.

Corresponding Author Name: Leo C James

Journal Submitted to: The EMBO Journal

Manuscript Number: EMBOJ-2018-101365R